# Implicit-ARAP: Efficient Handle-Guided Neural Field Deformation via Local Patch Meshing

**Daniele Baieri**
University of Milano-Bicocca
daniele.baieri@unimib.it

**Filippo Maggioli**
Pegaso University
filippo.maggioli@unipegaso.it

**Emanuele Rodolà**
Sapienza University of Rome
rodola@di.uniroma1.it

**Simone Melzi**
University of Milano-Bicocca
simone.melzi@unimib.it

**Zorah Lähner**
University of Bonn, Lamarr Institute
laehner@uni-bonn.de

## Abstract

Neural fields have emerged as a powerful representation for 3D geometry, enabling compact and continuous modeling of complex shapes. Despite their expressive power, manipulating neural fields in a controlled and accurate manner – particularly under spatial constraints – remains an open challenge, as existing approaches struggle to balance surface quality, robustness, and efficiency. We address this by introducing a novel method for handle-guided neural field deformation, which leverages discrete local surface representations to optimize the As-Rigid-As-Possible deformation energy. To this end, we propose the local patch mesh representation, which discretizes level sets of a neural signed distance field by projecting and deforming flat mesh patches guided solely by the SDF and its gradient. We conduct a comprehensive evaluation showing that our method consistently outperforms baselines in deformation quality, robustness, and computational efficiency. We also present experiments that motivate our choice of discretization over marching cubes. By bridging classical geometry processing and neural representations through local patch meshing, our work enables scalable, high-quality deformation of neural fields and paves the way for extending other geometric tasks to neural domains.

## 1 Introduction

Implicit representations—where a surface is defined not explicitly but, for example, as the zero level set of a signed distance function—have long been used in computer graphics. However, they have recently gained renewed attention, particularly due to advances in neural rendering techniques such as NeRF [36]. Neural fields provide a compact representation for implicit representations in the weights of a neural network, and offer several advantages: they support flexible topology, avoid predefined discretization, and integrate naturally with gradient-based optimization methods. These properties make neural fields well-suited for reconstruction tasks. As this representation becomes more widespread, the demand for tools that enable direct analysis and manipulation of implicit surfaces continues to grow.

The traditional representation used in geometry processing applications is the polygonal mesh, in which the surface is explicitly modelled by a collection of connected polygons. This representation

39th Conference on Neural Information Processing Systems (NeurIPS 2025).

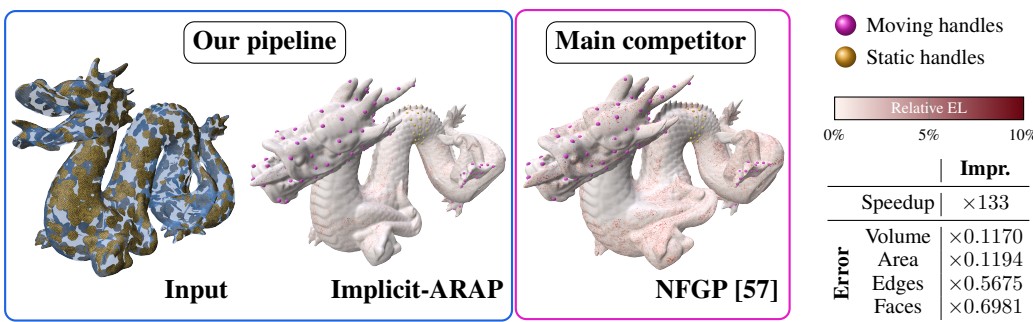

Figure 1: Given a neural surface and deformation handles, we employ our local patch mesh representation to optimize a neural deformation field via the ARAP loss. Our method introduces significant improvements over previous state of the art while employing $0.75\%$ of the time. We show the normalized change in edge lengths (before/after deformation) as a colormap on the deformed surfaces.

provides a high level of interpretability and allows for fine-grained manipulation by artists. Additionally, local surface properties can be easily computed because the neighborhood information is explicitly encoded. Due to its wide adoption and easy manipulation, a number of editing methods have been developed for explicit representations: one of them is optimizing the as-rigid-possible (ARAP) energy to deform an object while satisfying user-defined handle constraints and at the same time preserving the surface geometry (in terms of its edges) to a reasonable extent. This yields natural deformations and has been widely adapted in various applications [7, 37, 23].

Evaluating energies such as ARAP directly on implicit surfaces is challenging: properties like edge lengths cannot be computed without applying a discretization scheme, and equivalent properties do not necessarily exist for isosurfaces. Alternating between both representations is certainly possible [35], but it is expensive and inherits the sensitivity to discretization choices [57]. To counter this, we propose a new patch-based meshing for implicit representations which is 1) efficient to compute even at high resolution, 2) not sensitive to the specific choice of discretization, and 3) can be used on all isosurfaces to cover the complete geometric information of the neural field. Along with local patch meshing, our work proposes multiple other significant contributions:

- Extending the ARAP estimation of the thin-shell energy to neural representations;
- Providing an efficient method for handle-guided deformation of neural fields (see Figure 1);
- Introducing a robust and efficient alternative for deforming high-resolution meshes.

## 2 Related work

**Handle-Based Deformations**    Mesh-based shape editing methods have a long history in geometry processing due to the wide acceptance and explicit representation of meshes [58, 6]. Among these, methods which use handles to indicate the preferred deformation are intuitive for human users to understand and generate. The as-rigid-as-possible (ARAP) energy [51] has become popular due to its straight-forward interpretation and easy optimization while satisfying handle constraints. However, while not complex, its global nature still prevents processing of high-resolution shapes and is sensitive to the mesh discretization. The second aspect was overcome in SR-ARAP [26] with a smoothed and rotation-enhanced ARAP version. While the rigid version is widely used in variety of applications [7, 37, 23], there exist similar formulations other energies related to shape deformation, for example conformal [42, 55] and elastic deformation energies [9]. Instead of using a pre-defined physical energy, it is also possible to learn a set of handles from a collection of shapes [29, 41]. However, these require a suitable set of training data and are restricted to the space of deformations learned initially.

The idea of parametrizing shape deformations with neural networks has been previously explored in recent research: two contributions which are particularly close to our work are Neural Jacobian Fields [2] and Neural Shape Deformation Priors [53]. NFJ is a mesh morphing (*i.e.*, transfer of pose between two given shapes) model which can be trained to inject several different priors, including ARAP deformations. NSDP, on the other hand, is a data-driven handle-based mesh deformation

model. While it allows for very fast deformation of low/medium-resolution meshes, it is limited to the deformation prior and the handle set observed in the training data, limitations overcome by Implicit-ARAP.

While all these methods are designed for polygonal meshes, there are various types of shape representations for which other types of deformation approaches are better suited. One such example which is especially relevant to our work is spatial deformation tasks, which apply to solid models [46, 47], radial basis functions [5] or cages for character animation [24].

**Neural Fields Editing** Energies like ARAP act directly on surface deformations, making them naturally suited to explicit representations such as triangle meshes. In contrast, editing implicit surfaces is more challenging due to their global coupling – local changes to the surface can affect distant regions of the field. Even locating the surface in space can be non-trivial without certain assumptions. The recent popularity of NeRF-like methods [36] has renewed interest in integrating deformations into implicit frameworks. Dynamic NeRFs have been implemented by introducing a time parameter [45] or optimizing deformation fields [8]. Other works modify the MLP weights of neural fields to generate [17] or edit [3] shapes directly, though this lacks efficient formulations of explicit energies like ARAP and leads to slow optimization. Mehta et al. [35] alternate between explicit and implicit representations to enable deformations, but require costly conversions at each step. Novello et al. [39] supports diverse deformations but lacks handle-based constraints. Neural fields with spatial features allow shape editing through feature-space interpolation [20, 1]. Similarly to our approach, Esturo et al. [18] optimize all isosurfaces simultaneously using divergence-free fields – an approach also applicable beyond implicit functions [15]. Recent work [14] shows cage-based deformations can be modeled with neural fields by learning mappings to barycentric coordinates. Text-driven editing of implicit or hybrid representations (e.g., Gaussian splatting) has also been explored [30, 56, 19, 10, 40].

The method closest to ours, due to Yang et al. [57], optimizes deformation losses sampled from implicit representations but is highly inefficient, requiring several hours. In contrast, we leverage full neural field information efficiently by using small, randomly placed discrete patches. Our use of mesh-based ARAP objectives is supported by [11], which highlights issues with higher-order neural field derivatives, and [28], which advocates finite differences over analytical gradients.

Other efforts to simplify local SDF representations include enhancing marching cubes with Bézier patches [54] and composing surfaces from implicit primitives [31]. However, unlike our approach, these methods do not yield local representations that are both explicit and structurally simple.

## 3 Method

Our method employs a local patch model (Sec. 3.1) to sample the surface of a neural signed distance field, which is then deformed according to an ARAP-like energy (Sec. 3.2) with an efficient optimization scheme (Sec. 3.3).

### 3.1 Local patch meshing

Given a 3D surface $\mathcal{S}$ represented implicitly by a neural signed distance field $f_\theta \colon \mathbb{R}^3 \to \mathbb{R}$, we require a discrete local representation in order to compute the ARAP energy induced by a given deformation field $d \colon \mathbb{R}^3 \to \mathbb{R}^3$. We achieve this by generating local patches for *multiple* isosurfaces of $f_\theta$. The procedure is separated into sampling and projection:

**Sampling.** We start by sampling $k$ points $V = \{v_j\}_{j=1}^k$ from a disk with radius $\rho$ on the 2D plane (including the origin) and computing the planar Delaunay triangulation. There are several possible distributions to sample the 2D disk: we describe and visualize some possible options in the supplementary material. However, our experiments showed that the particular choice has little influence on our method, as we show in Figure 9.

**Projection.** Once we obtain the 2D patch $\mathcal{P} = (V, F)$, we can discretize each isosurface by projecting the patch and fitting it to the local geometry. To that end, we approximate a uniform sampling of $n$ points $O = \{o_i\}_{i=1}^n$ from the zero level set $\mathcal{S}_0$ using the rejection/projection algorithm proposed by Yang et al. [57]. Additionally, we sample points from other level sets by sampling

the 3D interval $[-1; 1]^3$ uniformly at random. We project a patch for each sampled point as $V_i = \{o_i + R_i\left(t\left(v_j\right)\right)\}_{j=1}^k$, where $t$ maps 2D points to 3D as $t \colon (x, y) \mapsto (x, y, 0)$, $o_i$ are the sampled origin points, and $R_i$ are the rotations aligning patch and surface normals at each $o_i$, which we obtain

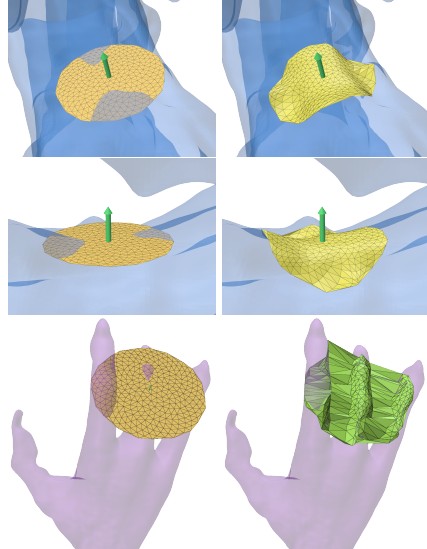

as the normalized gradient of $f_\theta$ (first column of Figure 2). Then, we map each patch vertex onto level set $l = f_\theta(o_i)$ via the SDF closest surface point formula [57, 12] which maps any 3D point to its nearest neighbor on level set $l$ as

$$p' = p - (f(p) - l)\frac{\nabla f(p)}{\|\nabla f(p)\|} . \tag{1}$$

In practice, when $f$ is a neural SDF, this formula has to be applied recursively to yield accurate results (*i.e.* points whose signed distance is approximately $l$). Note that the radius $\rho$ of the tangent disk specifies a measure of locality which is highly dependent on the underlying surface $\mathcal{S}$. Figure 2 shows examples of successful projection and radius overestimation.

By sampling origin points from multiple isosurfaces of the signed distance field $f_\theta$, we enable representing any local region of the neural field with a patch mesh. We describe how this structure can be applied to shape deformation in Section 3.3. Moreover, we compare it to the classic SDF meshing algorithm Marching Cubes [32, 33] in terms of its benefits for handle-guided deformation in Section 4.2.

Figure 2: Examples of patch projection in a favourable case (top two rows) and with overestimated radius.

## 3.2 Deformation model

Following previous work in neural field deformation [8, 39, 45, 57], we represent our deformation as a continuous function of the embedding space $d \colon \mathbb{R}^3 \to \mathbb{R}^3$. We model it via a MLP $g_\phi \colon \mathbb{R}^3 \to SO(3) \times \mathbb{R}^3$ mapping 3D coordinates to roto-translations, with parameters $\phi$. We use $R_\phi$ and $t_\phi$ to refer to the rotation and the translation fields separately. The output layer predicts a 6D vector, where the first three components are interpreted as Euler angles and converted to a rotation matrix. We define the complete deformation as $d_\phi(x) = (R_\phi(x) \cdot x) + t_\phi(x)$.

## 3.3 Optimization

We optimize our model similarly to Yang et al. [57]; we summarize the process in the supplementary material. The goal is to optimize for target handle positions while regularizing the computed deformation to have some desired properties. Given a set of handles as with source-target position pairs $H = \{(s_i, t_i)\}_{i=1}^h$ (where $s_i = t_i$ in the case of static handles), we fit it via a simple MSE loss $L_{\text{handle}} = \frac{1}{h}\sum_{i=1}^h \|d_\phi(s_i) - t_i\|^2$. The key part of our loss function is the As-Rigid-As-Possible (ARAP) energy introduced by Sorkine and Alexa [51]. While this formulation was previously adopted as a regularizer for generative neural models [16, 23], our work is the first to employ it in the setting of implicit geometry processing. We aim to ensure that our map $d_\phi$ deforms the level sets of $f_\theta$ as-rigidly-as-possible. We evaluate this in a Monte-Carlo fashion, by sampling a set of points $\{x_k\}_{k=1}^n$, where $m \le n$ points are sampled uniformly from the zero level set of $f_\theta$ and $n - m$ points

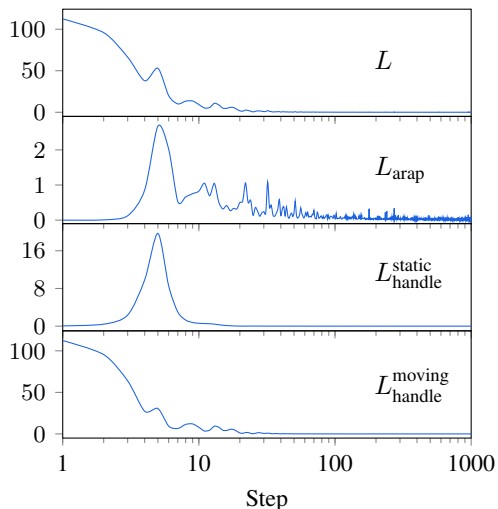

Figure 3: Loss curves for a single Implicit-ARAP training (Figure 9, first column). We show separate handle losses for static and moving handles for visualization purposes.

uniformly from the bounded volume $[-1; 1]^3$. A surface patch is computed at each of these points via the algorithm presented in Section 3.1, yielding a patch-based representation $\{(V_k, F)\}_{k=1}^n$. Then, the ARAP regularization is computed as

$$L_{\text{arap}} = \frac{1}{n} \sum_{k=1}^{n} \sum_{(v_i, v_j) \in E_k} w_{i,j} \left\| (d_\phi(v_i) - d_\phi(v_j)) - (R_\phi(v_i) \cdot (v_i - v_j)) \right\|^2 . \qquad (2)$$

Where $E_k$ are the mesh edges for a patch mesh $(V_k, F)$ and $w_{i,j}$ are the cotangent Laplacian edge weights. By optimizing the deformed edges (left hand side of the difference) to be as close as possible to the rotational part (right hand side), we effectively mitigate the action of the translation field. The original ARAP formulation only used vertex-wise rotations, as handles were fit as a pre-processing step via Laplacian smoothing of the handle function over the surface. This operation is non-trivial for implicit surfaces, therefore we include a translation field, which allows any given handles set to be fit. The network $g_\phi$ is optimized with ADAM [25] steps until convergence of the loss function $L = \lambda_1 L_{\text{handle}} + \lambda_2 L_{\text{arap}}$. The entire procedure for computing the $L_{\text{arap}}$ loss, which we described in this section, is repeated at each iteration, including all handle points as part of the surface sample. Nonetheless, we have observed convergence to be extremely quick, typically in the order of a few hundreds of iterations, as showed in Figure 3. In our experiments, we usually trained our model for a total of 1000 steps.

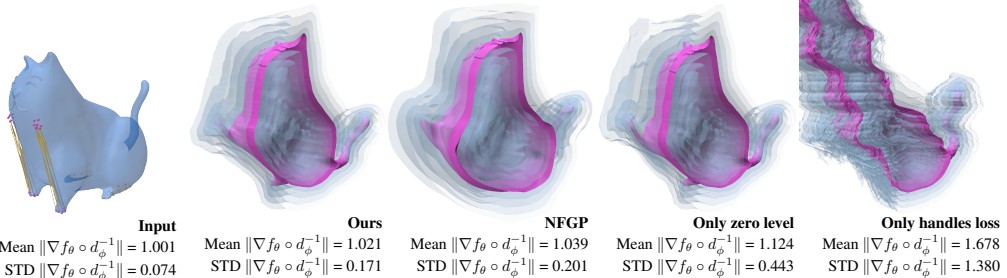

| Input | Ours | NFGP | Only zero level | Only handles loss |
|---|---|---|---|---|
| Mean $\|\nabla f_\theta \circ d_\phi^{-1}\| = 1.001$ | Mean $\|\nabla f_\theta \circ d_\phi^{-1}\| = 1.021$ | Mean $\|\nabla f_\theta \circ d_\phi^{-1}\| = 1.039$ | Mean $\|\nabla f_\theta \circ d_\phi^{-1}\| = 1.124$ | Mean $\|\nabla f_\theta \circ d_\phi^{-1}\| = 1.678$ |
| STD $\|\nabla f_\theta \circ d_\phi^{-1}\| = 0.074$ | STD $\|\nabla f_\theta \circ d_\phi^{-1}\| = 0.171$ | STD $\|\nabla f_\theta \circ d_\phi^{-1}\| = 0.201$ | STD $\|\nabla f_\theta \circ d_\phi^{-1}\| = 0.443$ | STD $\|\nabla f_\theta \circ d_\phi^{-1}\| = 1.380$ |

Figure 4: Ablation study for our method. We compare mean and standard deviation of the deformed SDF gradient norm for our method against: NFGP, our method with zero level set patches only, and our method without ARAP loss. The best results, in terms of both visual quality and preservation of the SDF field, are obtained by our full pipeline, proving the benefits of its individual components.

## 4    Experiments

The training procedure described in Section 3.3 can be applied seamlessly for both neural field and mesh deformation, by simply changing the MLP architecture (further details may be found in Section 6.2.1). We provide results for neural fields here, while we refer to Section 7.2 for an evaluation of high-resolution mesh deformation.

**Data and baselines**    We employ two datasets in our evaluation: the first one (TFD) is obtained by designing a set of hand-crafted deformation experiments using mesh data from Thingi10k [59] and the Stanford 3D scanning repository. The second one is the DeFAUST dataset introduced in [34]. We use this data to evaluate the performance of our method against several baselines. In our comparison, we consider the original As-Rigid-As-Possible deformation method proposed by Sorkine and Alexa [51] (ARAP). We also include the spokes and rims variant introduced by Chao et al. [9] (Elastic), who propose the inclusion of an elasticity model in the ARAP optimization, as well as the smooth rotations alternative (SR-ARAP) [26], where the ARAP energy is modified to improve smoothness and volume preservation. The last baseline is the neural field deformation approach introduced by Yang et al. [57], who propose to use the continuous formulation of the thin shell energy at random samplings of the implicit surface to optimize the deformation field, similarly to our method. For this baseline, we will indicate in the following whether the deformation is applied on the SDF field via its inverse (NFGP_SDF) or on the original input mesh (NFGP_Mesh). Although the Neural Shape Deformation Prior (NSDP) approach proposed by Tang et al. [53] achieves natural looking mesh deformations with interactive performance, it is a data-driven approach that requires the specification of consistent

| Input | NFGP$_{\text{SDF}}$ | **Ours$_{\text{SDF}}^{\text{Inv}}$** | ARAP | Elastic | SR-ARAP | **Ours$_{\text{SDF}}^{\text{MLP}}$** |
|---|---|---|---|---|---|---|

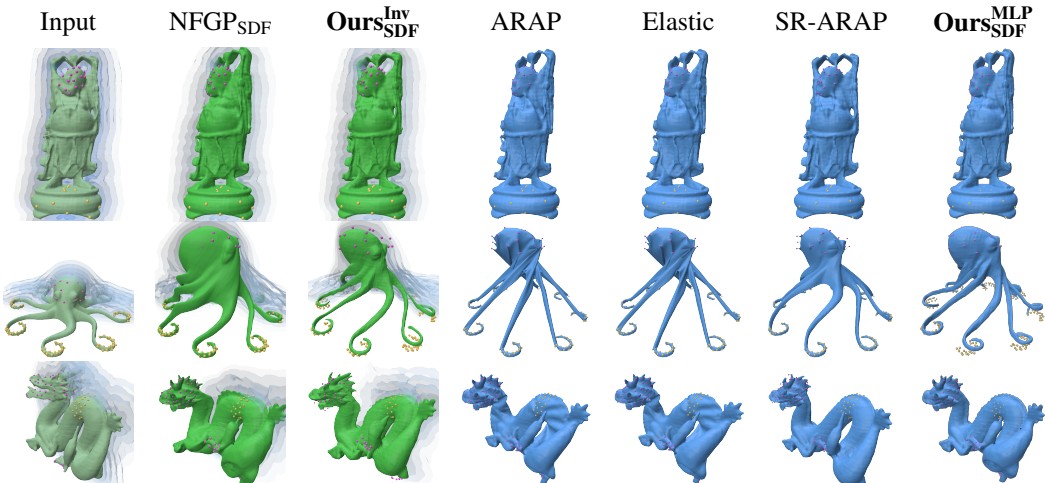

Figure 5: Qualitative results of our method in comparison to multiple baselines for neural field deformation. Discrete methods [51, 9, 26] are applied by extracting the marching cubes mesh at resolution $R = 512$ and applying a remeshing step, as suggested in [57]. We show true implicit deformations in green, while deformations of the zero-level set meshes are rendered in blue.

handles at training time; consequently, it is not suitable for a more general framework, and cannot be applied to our set of experiments. We conduct a separate comparison on mesh deformation between our method and NSDP in Section 7.3. Implicit-ARAP may be applied in multiple settings:

Ours$_{\text{Mesh}}^{\text{MLP}}$) regular MLP applied to input mesh, trained on its SDF representation

Ours$_{\text{SDF}}^{\text{Inv}}$) invertible MLP applied to input SDF field

Ours$_{\text{SDF}}^{\text{MLP}}$) regular MLP applied to the zero level set mesh of an input SDF

**Hardware and hyperparameters**  All of our experiments were run on a desktop computer with a 12GB NVIDIA RTX4070Ti GPU. Achieving efficiency in this setting allows us to show that our method is suitable for consumer grade hardware (and therefore end-user applications). The hyperparameters we used for the deformation tasks and the SDF fitting are listed in Section 7.1. We sample patch points via the sphere random uniform distribution (see Figure 10).

**Metrics**  To evaluate **meshing accuracy** with respect to some continuous implicit surface, we leverage the SDF $f_\theta$ and compare the respective level set value $l_i$ to $f_\theta(p)$ for surface points $p$ in the set of patches $\{\mathcal{P}_i\}_{i=1}^n$:

$$E_{\text{patch}} = \max_{i=1}^{n} \max_{p \in \mathcal{P}_i} \left| f_\theta(p) - l_i \right|. \tag{3}$$

In practice, the innermost max is estimated by evaluating the point-wise error for several points sampled from the triangles of the patch mesh $\mathcal{P}_i = (V_i, F)$. We chose to aggregate via maximum rather than mean, because a single outlier can create severe artifacts in the patch.

We use four metrics to quantitatively evaluate the computed deformations, considering both global and local aspects of the geometry. First, we consider the **percent error in volume and area** of the deformed geometry with respect to the original one. Given a surface $\mathcal{S}$ and its deformed version $\mathcal{S}'$, these are computed as

$$E_{\text{vol}} = \frac{\left| V_\mathcal{S} - V_{\mathcal{S}'} \right|}{V_\mathcal{S}}, \quad E_{\text{area}} = \frac{\left| A_\mathcal{S} - A_{\mathcal{S}'} \right|}{A_\mathcal{S}}, \tag{4}$$

where $V_\mathcal{X}$ and $A_\mathcal{X}$ indicate volume and surface area of shape $\mathcal{X}$. In order to evaluate the distortion induced on the input geometry, we use two distinct local criteria: **edge lengths and face angle errors**. To obtain consistent values across all experiments, we provide the former as a percentage of the

longest edge in the source mesh. Specifically, the error is computed as

$$\text{EL} = \frac{1}{|E|} \sum_{(u,v) \in E} \frac{|\|u - v\| - \|d(u) - d(v)\||}{\max_{e \in E} \|e\|} \ . \tag{5}$$

For the face angles, we compare the corresponding inner angles of source and deformed triangular faces:

$$\text{FA} = \frac{1}{3|F|} \sum_{f \in F} \sum_{(u,v) \in E(f)} \left| \cos^{-1} \left( \frac{u \cdot v}{\|u\|\|v\|} \right) - \cos^{-1} \left( \frac{d(u) \cdot d(v)}{\|d(u)\|\|d(v)\|} \right) \right| \ . \tag{6}$$

Both metrics require consistent connectivity between vertex sets. To implement them for the neural field pipeline, we extract the zero level set mesh from $f_\theta$ and forward-deform its vertices.

Table 1: Average runtime, angle error (FA) and percent errors in volume, area, and edge lengths (EL) for our neural field deformation experiments. Explicit ARAP baselines are run using the zero level set marching cubes mesh as input.

| | TFD | | | | | DeFAUST | | | | |
| | **Ours** | NFGP | ARAP | Elastic | SR-ARAP | **Ours** | NFGP | ARAP | Elastic | SR-ARAP |
|---|---|---|---|---|---|---|---|---|---|---|
| Volume | 0.00% | 6.66% | 9.40% | 9.01% | 5.60% | 0.00% | 15.15% | 17.36% | 17.33% | 12.09% |
| Area | 0.85% | 5.30% | 0.29% | 0.30% | 3.97% | 0.63% | 20.02% | 16.44% | 16.41% | 8.67% |
| EL | 3.26% | 3.34% | 0.33% | 0.38% | 3.44% | 2.41% | 8.71% | 5.72% | 5.73% | 8.45% |
| FA | 4.541° | 3.516° | 0.358° | 0.379° | 4.615° | 3.399° | 7.577° | 1.677° | 1.683° | 6.833° |
| Time | 2m:48s | 14h:26m | 9m:51s | 9m:44s | 10m:27s | 5m:36s | 14h:26m | 8m:11s | 8m:38s | 7m:06s |

## 4.1 Neural field deformation

The problem of deforming neural fields with handle guidance was first introduced by Yang et al. [57], but the literature is missing follow-up proposals of significant improvements over their work. Berzins et al. [3] hint at neural shape editing as one of the applications of their method, but a complete implementation is not available. Other baselines are obtained by applying explicit methods on the zero level set of a neural field, although these methods do not preserve the neural field information.

**Neural Field Pipeline**

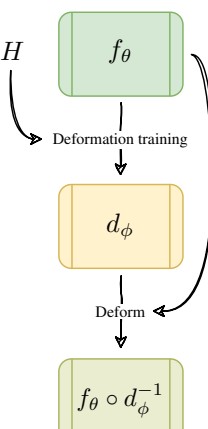

**Architecture** Following past literature [57, 38], we define the deformed field as $h(x) = f_\theta(d_\phi^{-1}(x))$. Intuitively, we obtain the SDF value by mapping the query point in the deformed space $x$ to its image $x'$ (s.t. $x = d_\phi(x')$) in the source space, via the inverse of the deformation. Therefore, we employ an invertible MLP architecture based on coordinate splitting, originally proposed by Cai et al. [8]. Since this architecture is derived from the NICE model [13], it retains its volume-preservation property, a notoriously useful prior for deformations (see Table 1). The network is composed of 6 coordinate splitting layers, where the individual coordinate processing blocks are implemented as 3-layer MLPs with Softplus activation, a hidden dimensionality of 256, and 6-frequencies Fourier features encoding [52]. We provide additional details about the invertible MLP architecture in Section 6.2.1.

**Results** We show averaged quantitative results for our method and baselines in Table 1. For true implicit methods (Ours and NFGP), we compute volume and area directly from the deformed field $h$. Since the EL and FA metrics require consistent connectivity between source and deformed shape, we compute them by deforming the zero level set mesh of $f_\theta$; the same holds for the discrete methods ARAP, Elastic and SR-ARAP. We used marching cubes resolution $R = 512$. From the data presented in the table, we observe that our method achieves optimal volume error due to the volume-preserving network architecture. Combining the results in Table 1 with the visualizations in Figure 5, we can appreciate how our method reliably yields plausible results in a fraction of the time required by the baselines, especially NFGP. Moreover, as the results over DeFAUST show, we can easily double the training iterations to achieve even better results in more challenging cases. The visible handle error in some of our outputs stems from our choice of loss balancing weights,

which we set to achieve the best results on average: if a more accurate handle fit is needed, the loss weights can be re-balanced, as we show in Section 7.6. Lastly, Figure 4 shows the benefits of both computing local patches for all level sets and employing the ARAP regularization for our deformation network. We use the deformed SDF gradient norm as a measure of preservation of the signed distance properties. We further discuss this aspect in Section 7.5.

## 4.2 Local patch meshing

The final section of our experiments is devoted to evaluating our local patch meshing algorithm. We begin by highlighting that **our method should not be considered as a drop-in replacement for marching cubes**: even by sampling a very large number of patches, it is unlikely to cover the entire surface, and a set of largely overlapping patches is not in general a useful representation for the surface. Instead, our method generates discretizations of local surface regions, and we are interested to a) verify how accurately these patches represent the underlying geometry and b) provide indications on how to select useful radius and density values for deformation tasks.

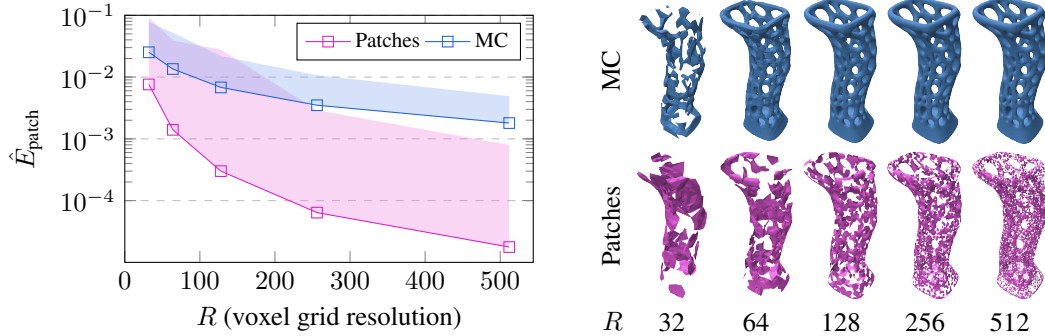

Figure 6: Accuracy evaluation of local patching against marching cubes. We construct the patch meshes in this experiment to match the total vertex count and average edge length of the corresponding marching cubes triangulation. The per-patch vertex count is fixed at 30 for all resolutions.

**Reconstruction Ability** In Figure 6, we compare marching cubes meshes to local patch meshes constructed with similar vertex count and average edge length. The visualizations show the change in "coarseness" of our representation wrt the resolution, while the graph provides some insights to the accuracy of local patch meshing. The line plot shows the *average* approximation error $\hat{E}_{\text{patch}} = \frac{1}{n} \sum_{i=1}^{n} \mathbb{E}_{p \in \mathcal{P}_i} |f_\theta(p) - l_i|$, with the shaded areas covering the entire region between $\hat{E}_{\text{patch}}$ and the maximum error $E_{\text{patch}}$ (Equation (3)). For MC, these metrics can be computed by considering the entire mesh as a patch (*i.e.*, $n = 1$ and $\mathcal{P}_1$ is the marching cubes mesh). Despite the marching cubes line hinting at a lower deviation from the mean, our method achieves much lower average error even for coarse patches. This is because the patch vertices are mapped exactly onto the surface, while marching cubes places triangles based on how the surface crosses the sampled voxels; therefore, our error only depends on the overall size of the patch relative to the "flat-ness" of the approximated local surface region.

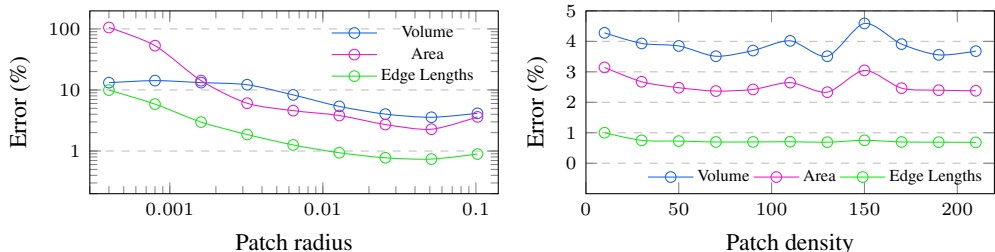

Figure 7: Variation of deformation error metrics wrt patch radius (left, fixed density $= 30$) and density (right, fixed radius $= 0.03$), averaged over TFD dataset.

**Approximation Error and Deformation Quality**
The inset figure (top right) shows how patch radius and density impact the approximation error (average over TFD shapes). We observe negligible change when varying the patch density, where values above 50 do not result in significant accuracy improvements. This is also the case for deformation error metrics (Figure 7, right). Given that increasing density results in additional cost in deformation time and memory (inset figure, bottom right), a conservative choice appears to be the best one. In our experiments, radius values around $0.01$ would usually provide geometrically meaningful patches without significant artifacts (we use absolute units for the radius since input shapes are normalized in the unit cube). On the other hand, while smaller radius values reduce the approximation error, the patches tend to collapse to single points as their radius approaches zero. This results in a less expressive representation of the local geometry, which negatively affects the deformation results as showed in the left plot of Figure 7. Lastly, overly large patches also cause a degradation in performance due to the increase in approximation error. In Section 7.8, we provide visualizations of Implicit-ARAP outputs as the patch radius and density change. The results are visually consistent with the quantitative evaluation presented here.

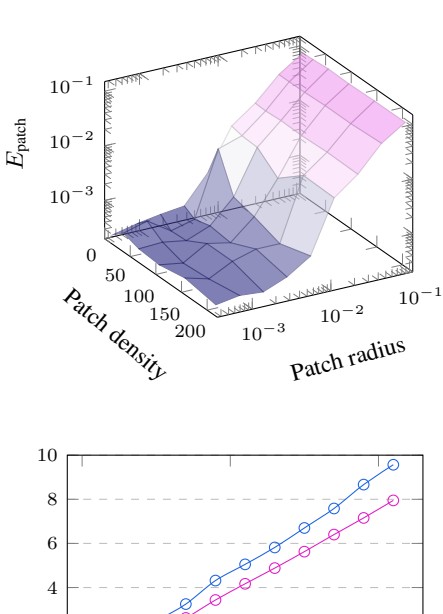

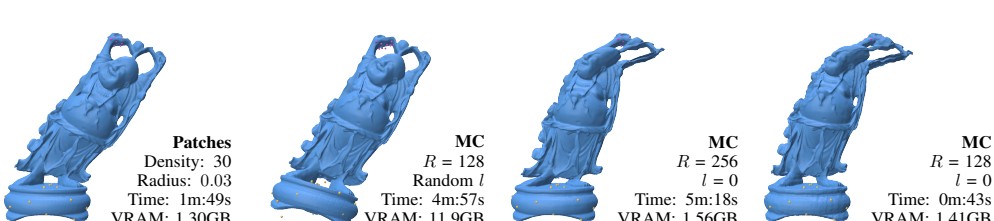

| **Patches** | **MC** | **MC** | **MC** |
| Density: 30 | $R = 128$ | $R = 256$ | $R = 128$ |
| Radius: 0.03 | Random $l$ | $l = 0$ | $l = 0$ |
| Time: 1m:49s | Time: 4m:57s | Time: 5m:18s | Time: 0m:43s |
| VRAM: 1.30GB | VRAM: 11.9GB | VRAM: 1.56GB | VRAM: 1.41GB |

Figure 8: LPM and MC as underlying triangulation for deformation tasks. Our sampling is both memory efficient and expressive.

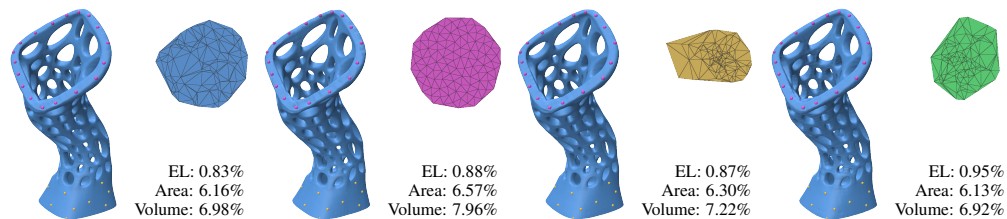

| EL: 0.83% | EL: 0.88% | EL: 0.87% | EL: 0.95% |
| Area: 6.16% | Area: 6.57% | Area: 6.30% | Area: 6.13% |
| Volume: 6.98% | Volume: 7.96% | Volume: 7.22% | Volume: 6.92% |

Figure 9: Mesh deformation results for multiple patch distributions. The final results are qualitatively identical and the variations in metrics are negligible in most cases. See Figure 6 for the input shape.

**Discretization** In Figure 8, we present qualitative results of mesh deformation using both patches and marching cubes (MC) as underlying discretizations for ARAP energy. We show three different variations for MC: zero level set only with resolution $R = 256$, zero level set only with resolution $R = 128$, and meshing all level sets with $R = 128$ to make the computed energy similar to that of our method. The last option leads to meshes with very high triangle count especially for higher SDF values, which increases the memory requirement and limits the resolution $R = 128$. Consequently, the optimization converges to a rigid transformation, probably due to the coarse-ness of the level sets'

representation. Using only the zero level set, the model seems unable to properly apply the ARAP prior to the deformation, for resolutions $R = 256$ and $R = 128$. This is likely due to using a single discretization during the optimization process: without seeing multiple possibilities (*e.g.* random patches), the continuous deformation model may find local optima that "cheat" the ARAP loss without actually resulting in local rigidity. Overall, our patching approach appears more stable, reliable, and efficient. Moreover, the results presented in Figure 9 suggest that the variation in error metrics due to the choice in patch point sampling is negligible.

## 5    Conclusions

**Discussion**    We presented a novel way to apply as-rigid-as-possible deformations to neural fields which is highly efficient and more flexible and robust than previous work. To this end, we proposed to mesh patches from several isosurfaces of a signed distance field and then compute the energy on those to regularize a deformation field encoded in a neural network. This has important advantages because it detaches the computational complexity from the resolution and allows for regularization that includes properties of the embedding space, *e.g.*, the volume-preservation of our invertible model. The core idea can be applied seamlessly in the context of deforming high resolution meshes and neural fields: in the latter case, we employ an invertible deformation which allows to define the output neural field, at the cost of generality. The combination of these properties – directly inferring the new SDF and general deformation space – is hard to obtain due to the unpredictable possible changes in the SDF from an unconstrained deformation, but it would make for a challenging future work. In the context of mesh deformation, we believe that employing more efficient neural SDF representations provides an interesting direction for future investigation. Nevertheless, we believe our work is a valuable step in the direction of efficient and flexible editing of neural fields, and that our local discretization could be applied to solve more geometric problems in the implicit domain.

**Limitations**    The general framework of our deformation method ("transporting" the input neural field along the inverse deformation) is adopted from previous contributions in the literature. As such, our method inherits the common limitation of lacking exact guarantees on the preservation of any properties of the input field (for instance, we discuss gradient norm preservation in Section 7.5). Additionally, the optimization of the ARAP loss is very dependent on the discretization: while our experiments showed that our patches are a valid choice, it would be beneficial to improve this representation with the goals of a) removing hyperparameters and b) increase surface coverage. Lastly, concerning mesh deformation (see Section 7.2), our method's constant time scaling is only beneficial for high resolution meshes: further improving the optimization time would allow our method to also target low/medium resolution meshes and improve its generality.

## Acknowledgments

This work was partially supported by the Italian Ministry of Education, Universities and Research under the grant *Dipartimenti di Eccellenza 2023-2027* of the Department of Informatics, Systems and Communication of the University of Milano-Bicocca and by the PRIN 2022 project *GEOPRIDE Geometric primitive fitting on 3D data for geometric analysis and 3D shapes*. We acknowledge the support of NVIDIA Corporation with the RTX A5000 GPUs granted through the Academic Hardware Grant Program to the University of Milano-Bicocca for the project *Learned representations for implicit binary operations on real-world 2D-3D data*. This work is further supported by the MUR FIS2 grant n. FIS-2023-00942 "NEXUS" (cup B53C25001030001), and by Sapienza University of Rome via the Seed of ERC grant "MINT.AI" (cup B83C25001040001).

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

# Contents

# 6 Additional details

## 6.1 Local patch meshing

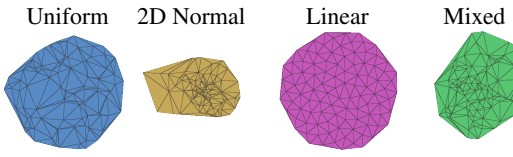

Figure 10: Different distributions for sampling the 2D disk (100 pts per patch).

We visualize four possible types of discretization in Figure 10, such as uniform sampling in polar coordinates, normalized normal sampling, linear (deterministic) sampling, and combining uniform coordinates with normal radius. Even though the patches are visibly different both in terms of point distribution and triangle appearance, the specific choice has little influence on our method, as we show in Figure 9.

Algorithm 1 presents the rejection/projection algorithm to sample the zero level set of an SDF introduced by Yang et al. [57]. In our implementation, several parts of the algorithm are parallelized for efficiency: for instance, the inner loop samples a large number of points $x$ at once, retaining and projecting those with absolute distance $< \tau$. By performing enough sampling attempts in a single iteration, the algorithm can frequently terminate in a single step (*i.e.*, by retaining at least $N$ points among those that were sampled).

---

**Algorithm 1** SDF zero level set sampling.

---

1: **procedure** REJECTPROJECTSAMPLING($f_\theta$, $N$, $\tau$, $t_{\max}$)
2:     $S \leftarrow \varnothing$
3:     **while** $|S| < N$ **do**
4:                                                                                     ▷ Rejection step
5:         $x \sim \mathcal{U}([-1; 1]^3)$                              ▷ Sample $x$ from bounded 3D domain
6:         **while** $|f_\theta(x)| > \tau$ **do**                              ▷ Ensure close to surface
7:             $x \sim \mathcal{U}([-1; 1]^3)$
8:         **end while**
9:                                                                                     ▷ Projection step
10:        **for** $t = 1 \to t_{\max}$ **do**                              ▷ Iterate closest surface point
11:            $x \leftarrow x - f_\theta(x)\dfrac{\nabla f_\theta(x)}{\|\nabla f_\theta(x)\|}$
12:        **end for**
13:        $S \leftarrow S \cup \{x\}$
14:    **end while**
15:    **return** $S$
16: **end procedure**

---

## 6.2 Model

### 6.2.1 Network architectures

**Shape model**    To represent the input shape internally to our deformation algorithm, we adopt a neural SDF model proposed in previous literature [50, 52]. This model is suitable to our application due to its efficiency on consumer-grade hardware and robustness with respect to the input geometry. The SDF is represented via a MLP network with 8 layers, a hidden dimensionality of 256, a residual connection at the fourth layer, 6-frequencies Fourier features encoding, and Softplus activation. This network is optimized via eikonal training, originally proposed by Gropp et al. [21], which employs the following four losses:

$$L_{\text{zero}} \quad = \mathbb{E}_{x \in \mathcal{S}_0} \left| f_\theta(x) \right|, \tag{7}$$

$$L_{\text{eikonal}} = \mathbb{E}_{x \in \mathbb{R}^3} \left\| \|\nabla f_\theta(x)\| - 1 \right\|^2, \tag{8}$$

$$L_{\text{normals}} = \mathbb{E}_{x \in \mathcal{S}_0} \left( 1 - \frac{\nabla f_\theta(x) \cdot \mathbf{n}(x)}{\|\nabla f_\theta(x)\|\|\mathbf{n}(x)\|} \right), \tag{9}$$

$$L_{\text{penalty}} = \mathbb{E}_{x \in \mathbb{R}^3} \exp\left(-\alpha \left| f_\theta(x) \right|\right). \tag{10}$$

Defined on a discretization of the represented surface $\mathcal{S}_0$. Intuitively, these respectively constrain the network to: **1)** vanish on surface points (sampled from the input mesh triangles) **2)** have unitary norm of gradient **3)** have gradient aligned with surface normals (indicated by $\mathbf{n}(x)$ for surface point $x$), and **4)** have minimal zero level set, to avoid artifacts due to under-determination. We list the values for loss weights and $\alpha$ which we employed in our implementation in Table 3. The Adam optimizer runs for a total of 10000 steps and uses a starting learning rate of $10^{-4}$ and a scheduler which halves it at steps 1000, 2000, and 5000.

**Deformation model.** For the invertible network employed in the neural field deformation pipeline, we use 6 coordinate splitting layers, where the individual coordinate processing blocks are implemented as 3-layer MLPs with Softplus activation, a hidden dimensionality of 256, and 6-frequencies Fourier features encoding [52]. Each of these layers splits and combines coordinates according to the layer index $i$, by selecting the "focus" coordinate $w = p_{i \bmod 3}$, where $p = (x, y, z)$ is the input vector. Each layer predicts a translation of the "focus" coordinate and a 2D roto-translation of the two others, which we progressively aggregate to obtain $R_\phi(x)$ and $t_\phi(x)$. The layer architecture for this model is visualized in Figure 11. Contrarily to the Lipschitz-continuous MLP used in NFGP [57], this architecture allows for an analytic expression of its inverse and thus is more efficient, as it does not require fixed point iterations for inversion.

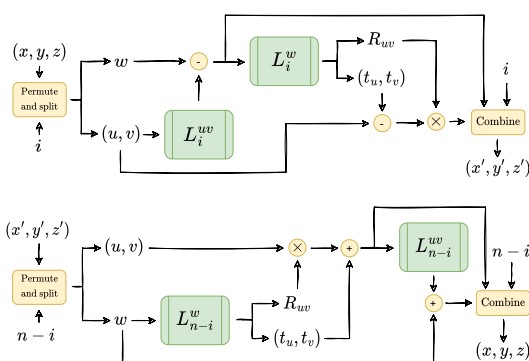

Figure 11: Diagram for the forward (top) and inverse (bottom) passes of the invertible MLP architecture we employ in our neural field deformation pipeline.

For the mesh deformation pipeline, we use a standard MLP composed of 8 linear layers with a hidden dimensionality of 256 and Softplus activation. We apply Fourier features encoding with 6 frequencies at the input layer and a residual connection at the 4th layer. For both networks, we adopt the neural deformation initialization scheme of [43, 8], which allows the initial state of the model to predict the identity transformation without causing symmetry breaking failures or numerical instability interacting with the Adam optimizer (which are common when the weight matrices are entirely initalized to zero).

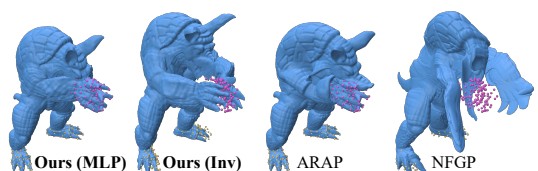

**Ours (MLP)** **Ours (Inv)** ARAP NFGP

Figure 12: Results for method and baselines for a non-bijective deformation, which invertible networks (Inv, NFGP) cannot represent.

## 6.3 Training procedure

Algorithm 2 shows all the high-level operations performed in a single step of Implicit-ARAP training. First, a single flat patch is constructed via the sampling procedure described in Section 3.1. Then, we construct a set of origin points for surface patches: we include 1) the handle sources, 2) a quasi-uniform sampling of the zero level set (see Algorithm 1) and 3) a uniform sampling of the bounded embedding space. The for loop in lines 7–10 aligns the flat patch at each origin and deforms it according to the local geometry. This operation is performed in parallel for all patches to avoid a bottleneck. Finally, the patch mesh is constructed as the union of all surface patches, and its vertices are deformed via $d_\phi$. The computation of the ARAP and handle losses is described in Section 3.3.

**Algorithm 2** Implicit-ARAP training loop.

---

1: **procedure** $\text{TRAIN}(T, \lambda_1, \lambda_2, n, m, \rho, k, f_\theta, d_\phi, \{(s_i, t_i)\}_{i=1}^h)$
2:     **for** $t \leftarrow 1$ **to** $T$ **do**
3:         $V, F \leftarrow \text{DISKSAMPLE}(\rho, k)$                               $\triangleright$ Section 3.1, "Sampling"
4:         $O \leftarrow \{s_i\}_{i=1}^h$                                     $\triangleright$ Handle sources as origins
5:         $O \leftarrow O \cup \text{REJECTIONSAMPLE}(f_\theta, m - h)$
6:         $O \leftarrow O \cup \text{UNIFORMSAMPLE}([-1; 1]^3, n - m)$
7:         **for** $j \leftarrow 1$ **to** $n$ **do**                             $\triangleright$ Section 3.1, "Projection"
8:             $V_j \leftarrow \text{ALIGN}(V, o_j, f_\theta)$
9:             $V_j \leftarrow \text{PROJECT}(V_j, f_\theta)$
10:         **end for**
11:         $\mathcal{P} \leftarrow \{(V_j, F)\}_{j=1}^n$
12:         $\mathcal{P}' \leftarrow \{(d_\phi(V_j), F)\}_{j=1}^n$                           $\triangleright$ Section 3.2
13:         $L_{\text{arap}} \leftarrow \text{ARAPLOSS}(\mathcal{P}, \mathcal{P}')$                       $\triangleright$ Section 3.3
14:         $L_{\text{handle}} \leftarrow \frac{1}{h} \sum_{i=1}^h \|d_\phi(s_i) - t_i\|^2$              $\triangleright$ Section 3.3
15:         $L \leftarrow \lambda_1 L_{\text{handle}} + \lambda_2 L_{\text{arap}}$
16:         $\text{OPTIMIZE}(L; \phi)$                             $\triangleright$ Compute gradients, ADAM step
17:     **end for**
18: **end procedure**

---

## 6.4 Implementation

We implemented our algorithm in Python, relying on PyTorch [44] for neural network primitives, linear algebra and automatic differentiation. In addition, we used Polyscope [49] for visualization, extending its GUI with functionalities for easy point picking, which we used to design deformation experiments. While our viewer renders a 3D triangle mesh extracted with marching cubes for the sake of efficiency, the points selected on the shape by the user are mapped exactly onto the implicit surface via iterations of the SDF closest point equation (Equation (1)), allowing to select an arbitrary set of accurate handles. For a given set of points, the user can then specify an affine transformation and save both the resulting handle transforms and the original positions. Our codebase is available at this url.

# 7 Additional experiments

## 7.1 Hyperparameters

Where unspecified, all of our deformation experiments were run using the hyperparameters showed in Table 2. To train the neural SDFs, we used the architecture described in Section 6.2.1 with the hyperparameters listed in Table 3.

Table 2: Hyperparameter values for the deformation procedures. $k$ and $\rho$ refer to patch density and radius, respectively, while $\lambda_i$ are the loss balancing weights. $n$ and $m$ define the number of sampled patches per training step (see Algorithm 2).

| | Mesh | | Field | |
|---|---|---|---|---|
| Data | TFD | DeFAUST | TFD | DeFAUST |
| Steps | 1000 | 2000 | 1000 | 2000 |
| LR | $10^{-3}$ | $10^{-3}$ | $10^{-3}$ | $10^{-3}$ |
| $\lambda_1$ | $10^3$ | $10^3$ | $10^3$ | $10^3$ |
| $\lambda_2$ | $10^1$ | $10^4$ | $10^3$ | $10^4$ |
| $k$ | 30 | 30 | 30 | 30 |
| $\rho$ | 0.03 | 0.01 | 0.03 | 0.01 |
| $n$ | 2000 | 2000 | 2000 | 2000 |
| $m$ | 1000 | 1000 | 1000 | 1000 |

Table 3: Hyperparameter values for the SDF fitting procedure.

| $\alpha$ | $\lambda_{\text{zero}}$ | $\lambda_{\text{eikonal}}$ | $\lambda_{\text{normals}}$ | $\lambda_{\text{penalty}}$ |
|---|---|---|---|---|
| 100 | 3000 | 100 | 50 | 3000 |

| Input | ARAP | Elastic | SR-ARAP | NFGP$_{\text{Mesh}}$ | **Ours$_{\text{Mesh}}^{\text{MLP}}$** |
|---|---|---|---|---|---|

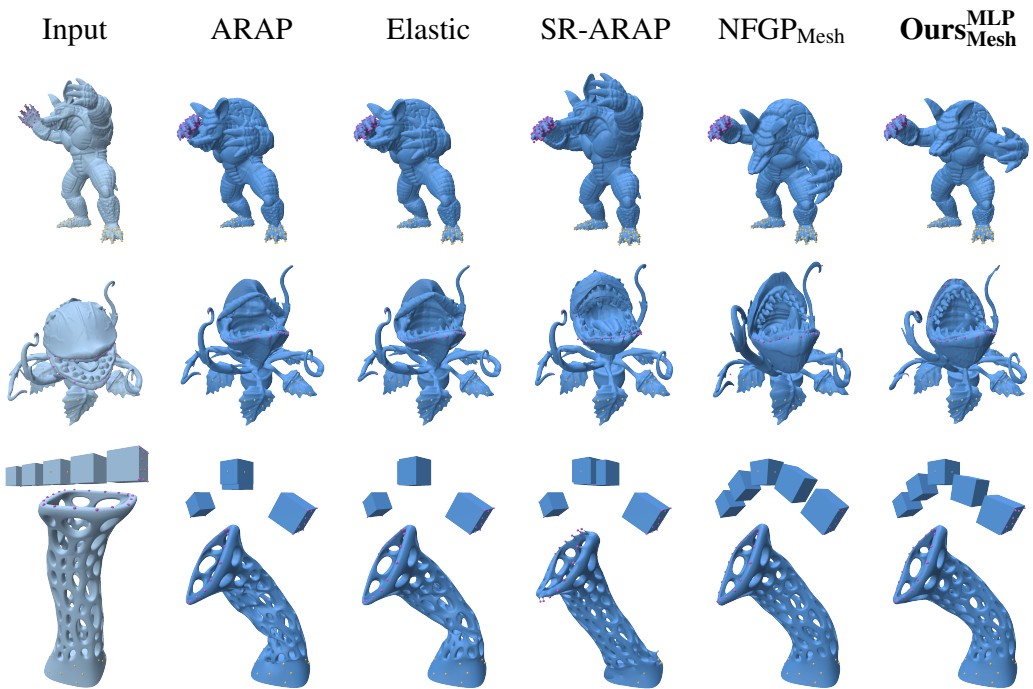

Figure 13: Qualitative results of our method in comparison to multiple baselines for mesh deformation, on TFD examples.

## 7.2 High resolution mesh deformation

Our pipeline can be applied for classic mesh deformation, where the trained network is used to deform the vertices of an input mesh (internally converted to a neural SDF for optimization).

**Architecture** For the mesh deformation pipeline, we do not require invertibility as we are simply interested in the forward deformation of a set of points in the zero level set of $f_\theta$. Therefore, we use a standard MLP composed of 8 linear layers with a hidden dimensionality of 256 and Softplus activation. We apply Fourier features encoding with 6 frequencies at the input layer and a residual connection at the 4th layer. As mentioned in the main manuscript, the network is again initialized to predict the identity transformation [43, 8]. This architecture is $\sim 1.53 \times$ more efficient to query than the invertible one, resulting in faster runtime for the deformation procedure (excluding the internal SDF fitting). Figure 12 is also related to this section, as it shows how the standard MLP we use in this task may represent deformations such that bijections between source and deformed shape are not possible (*e.g.* changes of topology).

**Mesh Pipeline**

$$\mathcal{M} = (V, F)$$

SDF reconstruction

$H$    $f_\theta$

Deformation training

$d_\phi$

Deform

$$\mathcal{M}' = (d_\phi(V), F)$$

**Results** Well-established explicit methods [51, 9, 26] are very efficient for most use-cases, but they do not scale to meshes with millions of vertices due to super-linear time and memory costs. By representing the input geometry implicitly in the weights of a neural network, and only computing Laplacian edge weights for small triangle patches, our method achieves runtime and VRAM usage independent of the input size.

As highlighted in Table 4, Implicit-ARAP's runtime is comparable to that of the explicit baselines. However, the time scaling of our method is constant, therefore it is expected to remain efficient even at resolutions higher than those employed in our evaluation. Moreover, most (~72%) of our runtime is spent fitting a neural SDF to the input mesh: employing a faster procedure for this step could greatly improve our performance. The remainder of Table 4 summarizes the performance of our method: Implicit-ARAP yields a clear improvement over SR-ARAP and NFGP. However, ARAP and Elastic achieve much better metrics than our method for area, EL and FA. Since these two methods provide the best solution (in a least-squares sense) for local rigidity of the given input mesh under the handle constraints, this is to be expected. However, including useful priors such as smoothness (SR-ARAP) immediately aligns the scale of error values to ours and NFGP.

Combining these results with the qualitative evaluation we present in Figures 13 and 14 provides a clear picture of the accuracy, robustness and efficiency of our method, which consistently yields accurate results with minimal artifacts. We point out that the results of ARAP baselines for the cubes experiment (Figure 13) are correct, as the 2nd and 4th cubes are unconstrained and may be mapped arbitrarily. Implicit-ARAP and NFGP, on the other hand, exploit the spectral bias of neural networks to propagate the handle maps smoothly over the whole domain. However, NFGP deforms the individual cubes into trapezoids more evidently than our method. The last advantage in using an implicit method like ours lies in its independence from the quality of the input shape's connectivity: for example, the CGAL implementation of explicit baselines failed to run using the buddha mesh (see Figure 8).

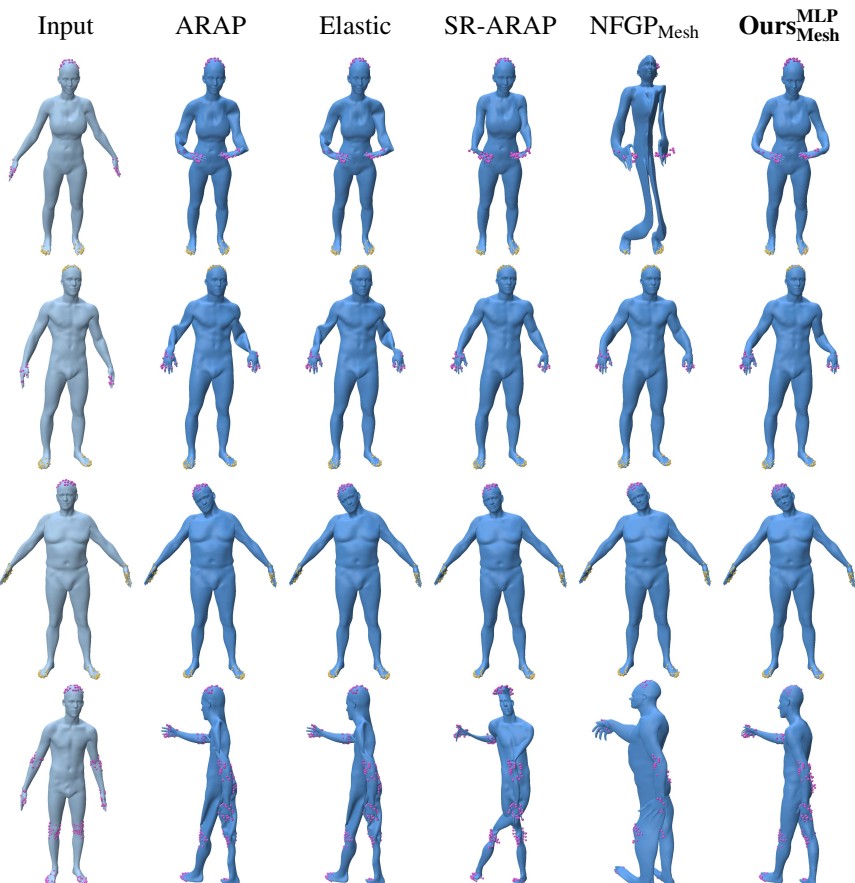

Figure 14: Qualitative results of our method in comparison to multiple baselines for mesh deformation, on DeFAUST examples.

Table 4: Average runtime, angle error (FA) and percent errors in volume, area, and edge lengths (EL) for our high-resolution mesh deformation experiments. The time for our method and NFGP includes 4m:44s required for the initial neural SDF fit, meaning our deformation phase runs in 1m:49s on average for 1000 steps.

| | TFD | | | | | DeFAUST | | | | |
| | **Ours** | NFGP | ARAP | Elastic | SR-ARAP | **Ours** | NFGP | ARAP | Elastic | SR-ARAP |
|---|---|---|---|---|---|---|---|---|---|---|
| Volume | 4.27% | 8.41% | 11.82% | 11.42% | 8.24% | 0.99% | 15.02% | 12.41% | 12.25% | 11.69% |
| Area | 2.83% | 7.04% | 0.23% | 0.25% | 3.46% | 0.82% | 20.25% | 0.45% | 0.45% | 5.36% |
| EL | 0.76% | 0.87% | 0.12% | 0.12% | 0.89% | 1.06% | 8.71% | 0.37% | 0.38% | 3.39% |
| FA | 3.448° | 4.178° | 0.486° | 0.495° | 4.743° | 1.517° | 8.859° | 0.517° | 0.523° | 5.628° |
| Time | 6m:33s | 14h:31m | 8m:06s | 8m:00s | 9m:07s | 8m:22s | 14h:31m | 7m:15s | 8m:27s | 7m:11s |

## 7.3 Comparison to NSDP

In this section, we compare Implicit-ARAP to NSDP [53], a recent contribution in neural methods for 3D mesh deformation. We highlight the main differences between the two methods in Table 6. Differently from Implicit-ARAP, NSDP is data-driven, therefore to ensure fairness we carry out this evaluation on the original test set employed by the authors, the DeformingThings4D dataset [27]. The results are available in Table 5. We observe that Implicit-ARAP, leveraging the local rigidity prior, achieves a clear advantage for all deformation quality metrics. However, on the low/medium resolution data available in DT4D, NSDP can compute deformations faster than Implicit-ARAP, by pre-training the model on large amounts of data. This comes at the cost of generality, as NSDP requires that the handles employed at inference time be "semantically" the same as those used during training (*e.g.*, limb extremities of a humanoid character).

Table 5: Comparison of Implicit-ARAP and NSDP [53] performance on DT4D [27].

| | Volume | Area | EL | FA | Time | VRAM |
|---|---|---|---|---|---|---|
| **Ours** | 0.58% | 2.04% | 2.03% | 4.728° | 6m:33s | 1.3GB |
| NSDP | 7.80% | 5.76% | 3.59% | 6.603° | 0m:18s | 2.1GB |

Table 6: Highlights of differences between our method and Neural Shape Deformation Priors [53].

| | **Implicit-ARAP** | NSDP |
|---|---|---|
| *Training* | Trained for each input | Pretrained on large dataset |
| *Handle set* | Any | Defined at **training** time |
| *Evaluation data resolution* | Medium/high (50k-500k) | Medium/low (10k-50k) |
| *Deformation prior* | Local rigidity (ARAP) | Linear blend skinning (data prior) |
| *Inference-time optimization* | Yes | No |

## 7.4 Ground truth evaluation

Throughout our evaluation, we ran deformation experiments which were obtained from 3D shape pairs (*i.e.*, shape pairs from FAUST [4], animations from DT4D [27]). In these cases, one may leverage the "target" shape information to evaluate how closely the algorithm approximates the deformation prior showed in the data, by computing the difference between the deformed shape and the target shape [53]. This provides an additional criterion of evaluation for deformation algorithms, which while not an "objective" metric of correctness, can provide additional insight into the behavior of a particular method. The results are available in Table 7.

Table 7: Evaluation of Implicit-ARAP's ability to approximate target shapes, when available for our deformation experiments. For reference, NSDP averaged $7.79 \cdot 10^{-4}$ on DT4D.

| DeFAUST | DT4D |
|---|---|
| $3.54 \cdot 10^{-3}$ | $3.44 \cdot 10^{-3}$ |

## 7.5    SDF preservation

We are interested in evaluating to what degree Implicit-ARAP deformations preserve the signed distance properties of the input field. Like past contributions, our method lacks theoretical guarantee of exact preservation, and obtaining such a guarantee is an open problem. In Figure 4, we provide an ablation experiment showing that our formulation is beneficial unitary gradient norm preservation. We complete that result with Table 8, which presents the average of the same data over all our experiments.

To give a practical measure of how well the SDF properties are preserved, we provide sphere-traced [22] renders of SDFs deformed with Implicit-ARAP in Figure 15. We used the sphere tracing implementation from Polyscope [49]. We note that, while ad-hoc rendering methods for deformed fields such as [48] could be used, the scope of this work allows us to be satisfied with these basic results. The results show that the deformed fields are rendered without significant artifacts.

Table 8: SDF gradient norm preservation, averaged over all our experiments. Only true implicit methods are evaluated. The mean and STD of the deformed field gradient norm are evaluated over a dense uniform sampling (1M points) of $[-1; 1]^3$. Recall that for exact SDFs $\|\nabla f\| = 1$.

|          | Mean $\|\nabla f_\theta \circ d_\phi^{-1}\|$ | STD $\|\nabla f_\theta \circ d_\phi^{-1}\|$ |
|----------|:----:|:----:|
| **Ours** | 1.016 | 0.146 |
| NFGP     | 1.029 | 0.290 |
| Input    | 0.999 | 0.077 |

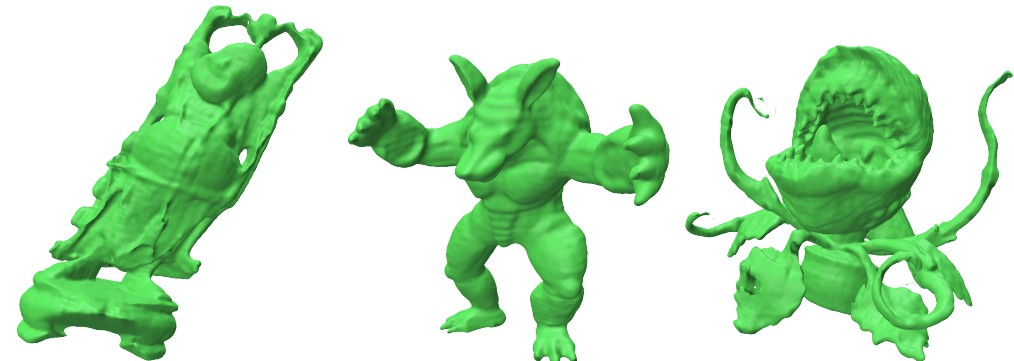

Figure 15: Sphere-traced renders of SDFs deformed with Implicit-ARAP. The deformation inputs may be found in Figures 5 and 13.

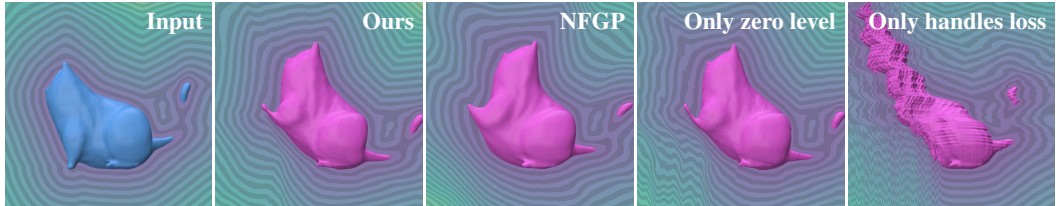

Figure 16: Alternative visualization of the ablation study presented in Figure 4.

## 7.6    Loss balancing

As we mentioned in the main manuscript, in our neural representation setting we cannot ensure a 100% accurate handle fit, as we have to optimize the input handles via gradient descent, balancing their contribution with that of the ARAP loss. As a result, especially in difficult cases with substantial deformations, the optimization may favor a better preservation of local rigidity and sacrifice some

measure of handle accuracy. In Figure 17, we show that it is possible to re-balance the loss term and achieve accurate handle fit, at the cost of a less natural-looking deformation.

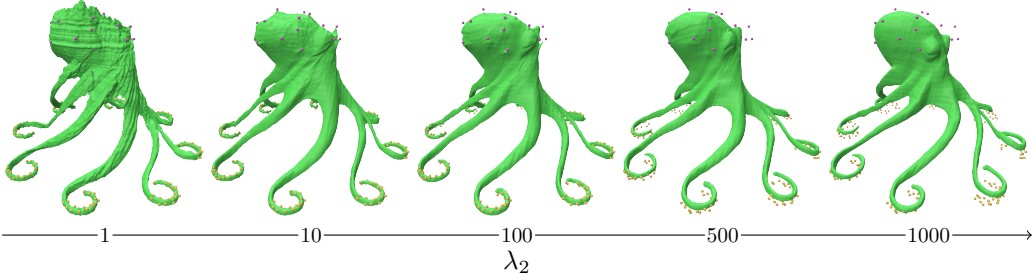

Figure 17: Decreasing the weight of the ARAP loss ($\lambda_2$) allows to balance handle satisfaction and deformation quality. $\lambda_1$ is set to 1000 as for all other experiments.

## 7.7 Matching ARAP resolution

Furthermore, we explore the connection between our method and the original ARAP through the results in Figure 18. ARAP results are generally sensitive to the input mesh resolution: different discretizations of the same shape will result in outputs which, while technically correct, exhibit different properties. For low resolution meshes, local rigidity will be optimized between large triangles, often resulting in behaviours typical of skeleton-based linear blend skinning deformations. On the other hand, high resolution meshes will exhibit local rigidity in more localized regions. This is apparent from the top row of Figure 18.

To determine if Implicit-ARAP would exhibit the same phenomenon, we ran our method varying radius and density so that the average edge length of our training patches would approximate that of the ARAP input meshes. Our results show that this is not the case, as the variations in our outputs are insubstantial.

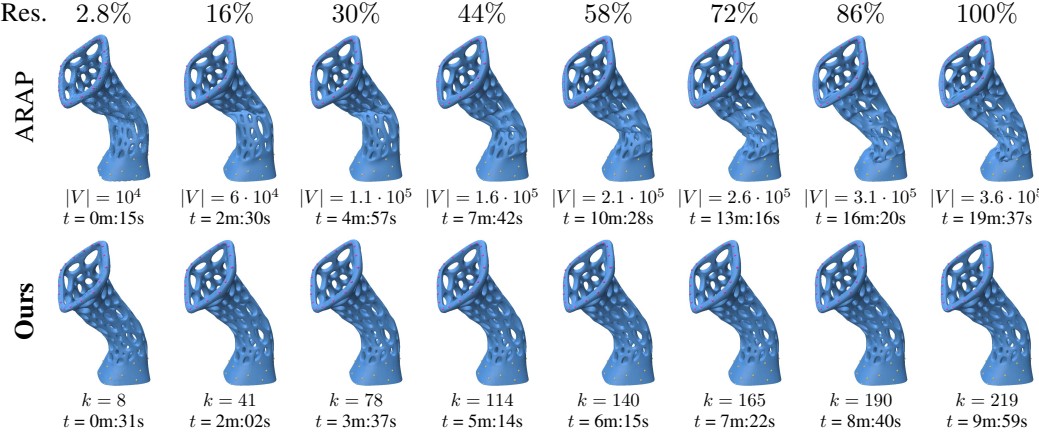

Figure 18: Comparison of our method to ARAP with matching resolutions (showed as a percentage of full resolution). The training patches are constructed with radius 0.03 (avoiding degenerate patches) and manually selecting the density $k$ which best approximates the average edge length of the corresponding ARAP input mesh (which we obtained by decimating the sculpture mesh at multiple resolutions). We also show the total runtime $t$ (without SDF fit in our case).

## 7.8 Correlation of deformation results and patch parameters

Lastly, Figure 19 completes the discussion about correlation of patch parameters and deformation performance Section 4.2. As we anticipated from the quantitative evaluation, we observe that small patches (radius below 0.01) result in artifacts due to the insufficient influence of the ARAP energy in

the optimized deformation. Viceversa, when the patches are too large, artifacts arise from interaction of geodesically distant regions (*e.g.*, the front left paw with the body, or the neck with the spine). Finally, coherently with the results in Figure 7, variations in patch density do not have a significant impact on visual quality.

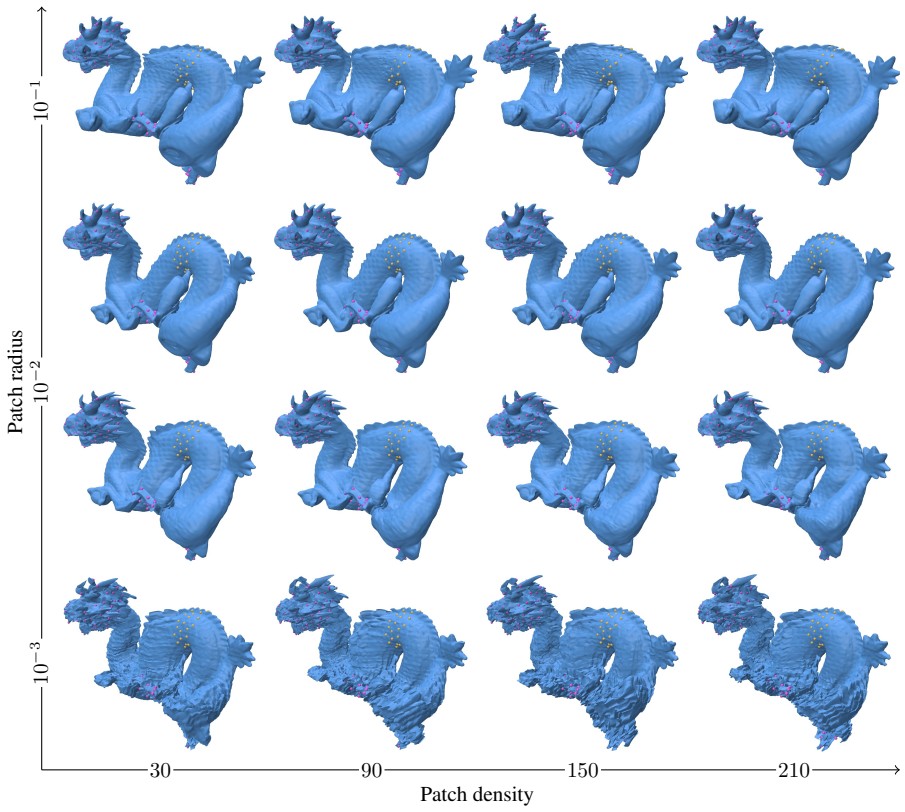

Figure 19: Qualitative results for Implicit-ARAP, varying patch density and radius.

## 8 Data license

For our experiments, we gathered a small collections of 3D shapes from Thingi10k [59] and the Stanford 3D scanning repository. Then, we designed various deformation experiments for each of these shapes. Finally, we included a set of deformation experiments automatically generated from the FAUST [4] dataset. In Table 9, we list the license name of each of these assets.

| Asset | Reference | License |
|---|---|---|
| DeFAUST shapes | Figure 14 | FAUST License |
| Armadillo (Stanford) | Figures 12, 13 and 15 | CC BY-NC 4.0 |
| Dragon (Stanford) | Figures 1, 2, 5 and 19 | CC BY-NC 4.0 |
| Buddha (Stanford) | Figures 5, 8 and 15 | CC BY-NC 4.0 |
| Piranha Plant (Thingi10k) | Figures 13 and 15 | CC BY-NC 4.0 |
| Hand (Thingi10k) | Figure 2 | CC BY-NC-SA 4.0 |
| Sculpture (Thingi10k) | Figures 6, 9, 13 and 18 | CC BY-NC 4.0 |
| Cat (Thingi10k) | Figure 4 | CC BY-NC-ND 4.0 |
| Octopus (Thingi10k) | Figures 5 and 17 | CC BY-SA 4.0 |

Table 9: References in the paper and license for all 3D meshes we employed in our experiments.

