# OpenReview forum: "Implicit-ARAP: Efficient Handle-Guided Neural Field Deformation via Local Patch Meshing"
_NeurIPS.cc/2025/Conference — NeurIPS 2025 poster_

### Official Review · Reviewer_BfYx · 2025-07-01

**Clarity:** 2
**Significance:** 1
**Originality:** 1
**Rating:** 4
**Confidence:** 3

**Summary:**

This paper presents a technique for handle-driven deformation of shapes represented as implicit surfaces. To enable plausible deformations, the authors propose a method for evaluating the widely used As-Rigid-As-Possible (ARAP) energy directly on implicit surfaces. Since ARAP traditionally relies on information such as edge lengths and surface connectivity—data that are not readily available in implicit representations—the paper introduces a strategy to approximate these quantities in the implicit domain. Specifically, the proposed method projects a set of planar meshes onto various level sets of the implicit function approximating the target surface. These projected meshes are then used to compute the ARAP regularization loss by leveraging the connectivity between their vertices. Combined with soft handle constraints, this regularization loss guides the optimization of a deformation field parameterized by a neural network.

**Questions:**

1. Have the authors considered using volumetric meshes when applying discretization-based deformation methods? Since the proposed method benefits from the volume preservation property of the NICE model, while many baselines operate solely on surface meshes, this comparison may seem somewhat unbalanced.
2. Could the authors provide statistics (e.g., mean and standard deviation) for the vertex and face counts of the meshes used in the comparisons? This includes both the raw outputs from Marching Cubes and their re-meshed versions.
3. While Table 1 reports that the proposed method achieves zero volume error, Figure 9 shows that all patch construction methods exhibit some degree of volume error. Could the authors clarify this discrepancy? Additionally, please provide dataset details—specifically, which shapes and how many were sourced from the Thingi10K and DFAUST datasets?
4. In the “Data and baselines” paragraph, the authors describe three variants of the proposed method. Which variant corresponds to “Ours” in Table 1?

**Ethical Concerns:**

["NO or VERY MINOR ethics concerns only"]

**Final Justification:**

The authors have addressed the concerns raised in my initial review by providing appropriate justifications and additional analyses. While the level of technical novelty appears modest, the proposed method—despite its simplicity—demonstrates clear performance improvements over the baselines. I also had concerns regarding the clarity and organization of the experimental section, but I believe these issues can be addressed in the final revision.

**Limitations:**

Yes. The authors discussed the limitations of their work.

**Paper Formatting Concerns:**

I find no noticeable formatting issue with this manuscript.

**Quality:**

2

**Strengths And Weaknesses:**

This paper presents a straightforward technique for estimating ARAP energy in the context of deforming implicit surfaces. The approach involves projecting a set of 2D mesh patches onto the target surface and leveraging their connectivity to approximate the ARAP energy.

While the core idea is interesting, I have some concerns regarding its practical applicability and use cases. A rich set of tools for processing implicit surfaces—such as mesh extraction, topology optimization, and mesh-based deformation—has been extensively studied and is already well-established in the computer graphics literature. As a result, while the paper demonstrates that deformation in the implicit domain is feasible, it remains unclear whether this approach offers tangible practical benefits over existing methods. In particular, the requirement to train and store a separate neural network for each deformation step could lead to inefficiencies, especially in interactive design scenarios where multiple deformations are applied in sequence. Addressing this limitation or proposing ways to mitigate it would further strengthen the work.

Although the paper targets a well-defined problem, the overall scope of the contribution feels relatively narrow—especially when compared to related efforts such as [1], which address a broader range of surface properties (e.g., normals, curvature) and additional applications like smoothing and sharpening.

Lastly, the experiment section feels somewhat unorganized. For example, acronyms such as “EL” and “FA” (Line 228) are used without prior introduction, and some results referenced in the text—such as those on the DFAUST dataset mentioned in Line 234—are not presented in the main paper. Clarifying these points would improve the clarity and completeness of the experimental evaluation.

[1] Geometry processing with neural fields, Yang et al., NeurIPS 2021

---

> ### Author Rebuttal · Authors · 2025-07-30
>
> We thank the reviewer for their constructive feedback about our work. We agree with the reviewer about the lack of clarity in some passages of the paper, which we attempt to clarify in this rebuttal.
>
> **Applicability and use cases**: the usage of neural fields in vision and graphics pipelines has grown more and more prominent in recent years, due to the massive efforts in this research area. One main limitation of these new representations is the lack of *native* tools for their manipulation and analysis: while most of them allow conversions to the mesh representation, this pattern nullifies the advantages of neural representations, such as unbounded resolution and independence from a particular discretization. We believe this motivates our research and several other efforts [1-6], solving problems directly in the neural field domain. Regarding interactive editing, the reviewer raises an interesting point: however, we did not propose interactive editing as a possible application because, while our method is ~130 times faster than the previous state of the art, it still is not fast enough for interactive usage. We believe further research in this direction would be extremely valuable, and that our work could be a starting point for such exploration.
>
> [1] Haque, Ayaan et al. "Instruct-NeRF2NeRF: Editing 3D Scenes with Instructions" Proceedings of the IEEE/CVF International Conference on Computer Vision. 2023.
>
> [2] Chen, Xu, et al. "Snarf: Differentiable forward skinning for animating non-rigid neural implicit shapes." Proceedings of the IEEE/CVF International Conference on Computer Vision. 2021.
>
> [3] Tao, Yuanyuan, et al. "Neural Implicit Reduced Fluid Simulation". SIGGRAPH Asia 2024 Conference Proceedings.
>
> [4] Yang, Lingchen, et al. "Implicit neural representation for physics-driven actuated soft bodies." ACM Transactions on Graphics (ToG) 41.4 (2022): 1-10.
>
> [5] Aigerman, Noam, et al. "Neural jacobian fields: Learning intrinsic mappings of arbitrary meshes". SIGGRAPH 2022 Conference Proceedings.
>
> [6] Tang, Jiapeng, et al. "Neural shape deformation priors." Advances in Neural Information Processing Systems 35 (2022): 17117-17132.
>
> **Additional applications**: ARAP is a fundamental tool in geometry processing (it also received the test of time award in SGP 2023). GPNF would provide important tools for geometry processing on neural fields, but ARAP-like deformations can only be obtained with GPNF in an unfeasible time scale (\~10hrs on a TitanX GPU). For this reason, we consider ARAP-like deformations a worthy task to which we specifically dedicated our effort. By contrast, the methods for smoothing and sharpening proposed in GPNF are already quite efficient (\~10mins on a TitanX GPU). Extending our method to include smoothing is extremely simple: once a patch is constructed and deformed, we may optimize the smoothness of the patch mesh directly. If the reviewer deems it necessary, we could include some qualitative results for smoothing experiments, which we expect to have more or less the same runtime as our deformation method.
>
> **Improving organization of experiments section**: we agree that the organization of this section could be improved. About the two particular points raised in the review:
> 1. The evaluation metrics were not described in the main paper due to space constraints. We refer to the supplementary material in L205 for a detailed description. However, for the camera-ready revision, we could move the full introduction of the employed metrics to the main text.
> 2. The DeFAUST results we reference in L234 are those in Table 1, linking to the previous sentence in the text. We will update the text to highlight this connection.
>
> **Figure 9 volume error**: we understand the confusion about this point. The deformation network employed in this experiment is a standard MLP (not volume preserving), like the one used for the experiments in Fig. 13 and 14. The motivation for this choice is that we aimed to evaluate the change in error metrics with respect to the choice of patch sampling, and the volume-preserving network would have prevented us from seeing the change in the volume error metric (as it would always have been 0%). We will clarify this in the camera-ready revision. The requested information about the experimental data is reported below.
>
> **Other questions**
>
> 1: To our knowledge, the baselines we employed in the evaluation of our method do not support volumetric meshes. Our method exploits the NICE properties to achieve volume-preserving deformation of surfaces.
>
> 2: The statistics for the resolution of the employed data are reported below.
>
> 4: We thank the reviewer for catching our mistake. The variant used in Table 1 is `Ours_SDF^Inv`, while the one used in Table 4 is `Ours_Mesh^MLP`.
>
>
> **Table R.1**: Statistics of evaluation data.
>
> | Dataset | Mean vertex count | STD vertex count | Mean face count | STD face count |
> |---------|-------------------|------------------|-----------------|----------------|
> |TFD original |197761|183709|395210|367031|
> |TFD marching cubes |181573|71397|363189|142839|
> |DeFAUST original |172850|13198|345697|26396|
> |DeFAUST marching cubes|277480|18649|554956|37299|
>
> **Data**: the DeFAUST dataset was introduced in [7], and it is obtained by employing sparse vertices of the FAUST shapes as deformation handles. The deformations are defined over a subset of per-subject shape pairs (same person in different poses) in the FAUST dataset, totaling 80 deformation experiments. The input shapes are also subdivided to achieve medium-high resolution.
> Since it is hand-crafted, the TFD dataset is of a smaller scale and counts 10 shapes, with 3 deformation experiments each. We list the shapes from Thingi10k and the Stanford 3D scanning repository below.
> * Buddha, https://graphics.stanford.edu/data/3Dscanrep/
> * Dragon, https://graphics.stanford.edu/data/3Dscanrep/
> * Armadillo, https://graphics.stanford.edu/data/3Dscanrep/
> * Piranha plant, https://www.thingiverse.com/thing:3625381
> * Octopus, https://www.thingiverse.com/thing:159217
> * Sculpture, https://www.thingiverse.com/thing:21126
> * Samurai, https://www.thingiverse.com/thing:6570837
> * Cat, https://www.thingiverse.com/thing:34965
> * Hand, https://www.thingiverse.com/thing:46891
> * Dino, from ARAP and NFGP experiments
>
> [7] Maggioli, Filippo et al. "SShaDe: scalable shape deformation via local representations". arXiv, 2024.

---

> > ### Comment · Reviewer_BfYx · 2025-08-07
> >
> > Dear Authors,
> >
> > I appreciate for preparing the responses to my previous queries. The concerns I previously raised have been addressed, and I will adjust my rating accordingly. I hope that the points discussed during the rebuttal period are reflected in the revised manuscript. In particular, I believe the paper can be further strengthened by improving the organization of Sec. 4—specifically by providing a clear overview of the baselines and metrics considered at the beginning of each subsection. This is especially relevant to Sec. 4.1, where the experimental setup is introduced at the start of Sec. 4, although most of the details are relevant only to Sec. 4.1 and not to 4.2.
> >
> > As a minor point, I would like to note that, to the best of my knowledge, both Neural Jacobian Fields (Aigerman et al., ACM TOG 2022) and NSDP (Tang et al., NeurIPS 2023), mentioned in the rebuttal, parameterize the deformation of explicit meshes using implicit neural fields, but do not directly manipulate the implicit fields themselves. Please ensure these works are accurately cited and appropriately discussed in the related work section of the revised manuscript.

---

### Official Review · Reviewer_A7SK · 2025-07-02

**Clarity:** 3
**Significance:** 3
**Originality:** 3
**Rating:** 5
**Confidence:** 4

**Summary:**

This paper proposes a local patch meshing strategy as a bridge between neural implicit fields and mesh-based shape manipulation. Instead of relying on explicit iso-surface extraction (e.g., via Marching Cubes), the method projects and deforms predefined patch meshes using signed distance function (SDF) values and gradients. By randomly sampling patch meshes from various level sets of a neural SDF, the approach enables the use of mesh-based deformation priors such as ARAP loss for shape manipulation.
They show reliable and efficient mesh deformations, offering better computational efficiency compared to full-surface extraction and remeshing pipelines. In addition, the paper includes extensive ablation studies on hyper-parameters such as patch size, density, sampling strategies.

**Questions:**

The current mesh deformation evaluation primarily focuses on geometric preservation metrics such as edge length, volume, and face angle. However, it might be beneficial to include vertex-wise error metrics between the deformed and target mesh, especially if temporal mesh sequences are available. In such cases, a subset of vertex displacements could serve as handle movements to be used as input constraints for the deformation method.
The paper could be further strengthened by including comparisons against two more recent methods,  Neural Jacobian Fields (NJF) and Neural Shape Deformation Priors (NSDP), which addresses similar challenges in non-rigid deformation.

**Ethical Concerns:**

["NO or VERY MINOR ethics concerns only"]

**Final Justification:**

Thanks for rebuttal and response. Most of my concerns are addressed.  I keep my score of Accept 5.

**Limitations:**

Yes

**Quality:**

3

**Strengths And Weaknesses:**

Strengths:
The work is well-motivated and presents an interesting idea by introducing ARAP-based deformation constraints into neural field representations, without requiring explicit mesh extraction. The proposed local patch meshing strategy provides a flexible and generalizable way to represent local surface geometry across arbitrary level sets of the SDF. It also demonstrates robustness to variations in patch structure, patch density, and input mesh resolution.
Figures 13 and 14 show that the method produces more plausible and stable mesh deformations on the TDF and DFAUST datasets compared to existing baselines, highlighting the practical advantages of the approach in real-world scenarios.

Weaknesses:
1)	From the quantitative results in Tables 1 and 4, the method is inferior to some baselines regarding Area, EL, and FA.
2)	Missing comparison against two recent works:
neural network optimization-based method -- NJF
learning-based shape manipulation method -- NSDP
Aigerman, et al. Neural Jacobian Fields: Learning Intrinsic Mappings of Arbitrary Meshes. SIGGRPAH 2022
Tang, et al. Neural Shape Deformation Priors. NeurIPS 2022

---

> ### Author Rebuttal · Authors · 2025-07-30
>
> We thank the reviewer for their positive and constructive comments about our work. We individually address each of the points raised in the review:
>
> **Missing baselines**: we thank the reviewer for pointing out NSDP and NJF as additional possible baselines for our evaluation. We will include both in our related work. Regarding the comparison to these two, we remark that:
> 1. NSDP’s architecture is bound to the handles used for training. In particular, the pretrained model released by the authors was trained on human/animal data with a fixed set of handles at the tip of each limb and on top of the head. This is a limitation of NSDP, which prevents us from evaluating it even on TFD and DeFAUST data, since the deformations in our datasets have variable amounts (and semantics) of handles. Therefore, the best option for a fair comparison is to evaluate Implicit-ARAP on NSDP’s test data separately from the other baselines.
> 2. NFJ does not support handle-guided deformation as a task, and we think it would not be possible to do an even comparison: as a morphing model, its “interface” requires two shapes A and B, to output the morphing of A into B. While it is true that the NFJ model can be trained to represent an ARAP deformation space, at inference time, it requires the pre-deformed shape to yield its output, and there is no way to condition the model’s output on the handles alone. For our evaluation, we require methods which take a shape and a set of handles and output the deformed shape, and including NFJ would result in an inconsistent comparison.
>
> Due to these reasons, we did not consider a comparison in the setting proposed in our evaluation. We will include this discussion in the paper. The table below shows an evaluation of Implicit-ARAP and NSDP: the two methods are evaluated on DeformingThings4D, the dataset used by NSDP for training, using the test split provided in the NSDP codebase. We combined unseen identities and unseen motions to construct the test set. The 3D shapes contained in this dataset range from \~10k to \~50k vertices. We will include these results, as well as some visualizations, in the camera-ready revision.
>
> |    | Volume  | Area | EL  | FA   | Time | VRAM |
> |----|---------|------|-----|------|------|------|
> |**Ours**|**0.58%**    |**2.04%** |**2.03%**|**4.728°**|6m:33s|**1.3GB** |
> |NSDP|7.80%    |5.76% |3.59%|6.603°|**.18s**  |2.1GB |
>
> For completeness, we also provide the following table, highlighting differences between our method and NSDP.
>
> |    | Ours  | NSDP |
> |----|---------|-----|
> |Handles | Any       | Defined at training time     |
> |Evaluation data resolution| Medium/high (50k-500k) | Medium/low (10k-50k) |
> |Deformation prior | Local rigidity | Data-driven (LBS animations) |
> |Training | Trained for each input | Pretrained on large dataset |
> |Inference-time optimization| Yes | No |
>
> **Performance gap in quantitative evaluation**: it is true that in some cases our method performs worse than mesh-based alternatives for the three metrics listed by the reviewer. This is motivated by the ARAP baselines using the same set of edges for optimization and metrics evaluation. This is not the case for our method and NFGP, which have no notion of the input discretization. In the neural field setting, our method still achieves results orders of magnitude better than the state of the art in the neural field domain (NFGP).
>
> **Vertex-wise error metrics**: we agree with the reviewer that some form of ground truth would be extremely helpful in evaluating deformation methods. However, even if temporal mesh sequences were available, using the vertex positions as “objective” would be incorrect, as their variation depends on the particular priors of the employed animation (such as linear blend skinning vs elastic simulations). While it is interesting to evaluate how closely a deformation method approximates (e.g.) LBS, it is not a metric of its performance. Furthermore, the data we employed does not have temporal mesh sequences, as the deformation examples in DeFAUST are obtained from FAUST, which only consists of a few poses of the subjects without any temporal or sequential relation.

---

> > ### Comment · Reviewer_A7SK · 2025-08-06
> >
> > Thanks for the response.
> >
> > The explaination of 'performance gap between ARAP and the proposed method in quantitative evaluation' looks fair for me, considering the ARAP used edge-based regularization during training.
> >
> > The authors provided additional comparisions against NSDP, and showed they can outperform NSDP in the listed metrics.
> > Regarding the vertex error metrics, it is still useful and meaningful to evaluate shape deformations using GTs, while handle-based manipulation is an ambiguous problem (one-to-many mapping issue).  It can still reflect whether the deformed shape is deviated from the possible GT.  Also, learning-based methods can approximate various deformation patterns, including and beyond LBS, elastic simulation, non-linear motions, and so on. It depends on your training data.
> >
> > For the D-FAUST dataset, it has temporal mesh sequences.

---

> > > ### Author Response · Authors · 2025-08-07
> > >
> > > We thank the reviewer for addressing our rebuttal.
> > >
> > > We agree that our task is one-to-many, which is the main motivation behind our evaluation setting. However, GTs evaluation could give us an idea about deviation from a given GT deformation (even if it is not the unique correct results). We highlight that NSDP was trained on LBS animations, which is a good reason to evaluate that method on its approximation of LBS animations. If the GT deformation is generated exploiting the LBS technique, then it is reasonable to expect that NSDP will produce more coherent results. This is not the case for our method, which does not have a data prior.
> > >
> > > If requested, we will be happy to add the GT evaluation of our method on DT4D (as it is the only one where we can compare our results to NSDP). Lastly we clarify that our dataset is DeFAUST and not D-FAUST. It was proposed in [1] as a dataset of deformation experiments obtained from FAUST shape pairs.
> > >
> > > [1] Maggioli, Filippo et al. "SShaDe: scalable shape deformation via local representations". arXiv, 2024.

---

> ### Comment · Reviewer_A7SK · 2025-08-08
>
> Thanks for your clarification. Most of my concerns are addressed. I will take your response into account when I update my final rating.
>
> I still believe that  we can get more insights about optimized/learnt deformation behaviours by comparing it against potential ground truths. While DeFAUST doesn't have the temporal sequences, you can still peform GT evaluation, as long as it has different deformation states of a same shape.  For example, A, B are from the same identity with different deformations. Given A and target handles from B as inputs, you can get B'. Then, you can compare B' against B. It would be great if you can add this in the final version.

---

> > ### Author Response · Authors · 2025-08-08
> >
> > We sincerely thank the reviewer for engaging in the discussion.
> >
> > We understand the importance of evaluating our method's outputs against a given GT in providing useful insight into our optimized deformations. We will run this experiment over the DeFAUST dataset and report our findings in the supplementary materials of the final revision, along with a discussion of these results.

---

### Official Review · Reviewer_YEpM · 2025-07-02

**Clarity:** 3
**Significance:** 3
**Originality:** 3
**Rating:** 5
**Confidence:** 3

**Summary:**

This paper address the existing challenge of deforming shapes represented by neural fields. It leverages local mesh representation to be deformed with guidance by the SDF and its gradient. Experimental comparison with baselines demonstrates the superiority in terms of deformation quality, robustness, and computational efficiency.

**Questions:**

- is it possible to approximate the deformed SDF with a new SDF, rather than composing the R and f with two separated models?

**Ethical Concerns:**

["NO or VERY MINOR ethics concerns only"]

**Final Justification:**

My original concerns are minor. The authors provided responses on those points. I will maintain my positive rating.

**Limitations:**

There is paragraph discussing limitations in the conclusion.

**Quality:**

3

**Strengths And Weaknesses:**

Strengths

- The technique is well developed. It leverages the ARAP in an elegant way to solve the deformation of SDF. The deformed shape can be represented as a invertible MLP of the deformation along with the original sign function.
- Experimental results study the robustness of the technique to the samples local patch.
- The comparison against baselines shows superiority.

Weaknesses
- The Fig.5 is challenging to look at, especially for the layers, although it delivers the information that the proposed method is better.

---

> ### Author Rebuttal · Authors · 2025-07-30
>
> We thank the reviewer for their positive comments on our work and are glad to see it was well received. We address issues raised by the reviewer individually:
>
> **Figure 5 presentation**: we agree with the reviewer that the presentation of the figure is not optimal. Unfortunately, it is somewhat complex to show the level sets of a neural field in a compact way; one possible alternative is to show a single slice of the non-zero level sets as a background render, overlapped with the 3D shape of the zero level set. Since this alternative limits visibility of the zero level set, we preferred the visualization currently used in the paper. If the reviewer deems it necessary, we can update the visualization for the camera-ready revision.
>
> **New SDF after deformation**: this is an interesting question that deserves further exploration. In general, our method does not provide a new SDF “out of the box”: a trivial way to obtain one would be to deform the zero level set and optimize a new SDF from scratch. However, this could easily be done using any other deformation method, meaning it is not a particular advantage of our pipeline. To our knowledge, there are no deformation methods that edit the neural SDF parameters to satisfy some deformation constraints while maintaining SDF properties. We thank the reviewer for highlighting this point, which could be an interesting direction for future research.

---

> > ### Comment · Reviewer_YEpM · 2025-08-01
> >
> > Thank the authors for providing the response.
> >
> > Fig. 5. It would be great to have further illustration figures in the supplementary material while keeping the current fig. 5 in the main text. If readers need to a detailed explanation of the figure, they can refer to the supplementary material to do so.
> >
> > New SDF. Adding a discussion in the paper would make sense to me. This is not necessary to be part of the work to be done in this paper.

---

### Official Review · Reviewer_GZaa · 2025-07-03

**Clarity:** 3
**Significance:** 3
**Originality:** 3
**Rating:** 4
**Confidence:** 4

**Summary:**

The paper proposes a local patch based meshing technique for the purpose of neural field deformation. It is positioned in contrast to Geometry Processing for Neural Fields (GPNF) paper in the sense that it is solving a more specific problem of handle guided neural deformation -- which is does in a fraction of time (and with higher quality) compared to GPNF. The local patch based meshing avoids the need for repeated marching cubes steps that make optimization very costly. Moreover the local mesh structure naturally provides the topology information to apply ARAP which is not possible in a pure neural field setting. Experiments are conducted large scale datasets such as Thingi10K and DeFAUST which corroborate the paper claims. Ablations are provided for various design choices made in the method which help in understanding contributions of individual components.

**Questions:**

1. Given that the paper is solving a specific part of the larger geometry processing problems handled by GPNF, I am curious to know what prevents the local patch based meshing to handle tasks other than ARAP? Perhaps a detailed discussion on this would be helpful to the reader.

2. Can you throw some clarity on why GPNF takes 14hrs?

**Ethical Concerns:**

["NO or VERY MINOR ethics concerns only"]

**Final Justification:**

The authors have provided sufficient clarity to my questions. Moreover the comments made by other reviewers have also been answered satisfactorily. Overall I find this paper a useful contribution and hence I vote an accept (score 5).

**Limitations:**

yes

**Quality:**

3

**Strengths And Weaknesses:**

Strengths:

1. For the task of ARAP, the approach taken by the paper makes total sense as local patching provides a natural means to apply ARAP loss. I am also hopeful that this approach can be explored in future works for other geometry processing/ graphics tasks like UV mapping, etc.

2. The performance gains achieved viz time saved and improved accuracy (especially in volume preservation) are noteworthy. I find the ablations quite useful.

Weakness:

1. The difference in method is not very clear with respect to GPNF. I find several similarities -- GPNF too seems to not require MC for its optimization. It looks like the use of an invertible neural network model for approximating the deformation field is also present in GPNF. So what are key differentiations other than the fact that a local mesh structure allows for ARAP like losses to be applied?

2. I don't get why GFNP's volume loss is also not zero given that it too uses an invertible network which in theory should also be volume preserving as is the case in this paper?

3. I am also not very clear about the reason for ARAP (mesh based) to have such artefacts as shown in Figure 4. Why is the paper theoretically superior in approximating deformations? Is this supposed to do with the discretisation of ARAP?

4. I believe the paper would benefit from giving a more gentle introduction to ARAP, and the baselines considered while highlighting the key differences.

---

> ### Author Rebuttal · Authors · 2025-07-30
>
> We thank the reviewer for their positive and constructive feedback about our work. We answer each point individually:
>
> **Application to other geometry processing tasks**: in this paper, we decided to target shape deformation specifically, to address the lack of a method solving this relevant task in a feasible time scale. There is no obstacle in applying our local patch meshing method to the other simpler tasks addressed in GPNF (smoothing and sharpening); however, the solutions proposed by the authors for these other applications are already quite efficient (~10 minutes on a TitanX GPU). Additionally, our general “framework” (constructing local patch meshes > neural deformation > optimizing the deformation by some property of the local geometry) could adapt naturally to several other tasks (e.g. UV mapping).
>
> **Differences to GPNF + GPNF inefficiency**: we agree that in the paper, we did not provide extensive details about GPNF as a deformation method. We provide them here, and, given the opportunity, we will improve this part in the camera-ready revision. The overall framework of GPNF is similar to ours, i.e., a neural deformation is applied to a neural field and optimized while leaving the shape’s parameters untouched. However, the authors of GPNF chose to formulate the shape deformation problem in a way that is coherent with the continuous structure of neural fields, directly optimizing elastic energy (stretching and bending terms) at different sampling of the zero level set throughout the optimization. These energy terms depend on 2nd degree derivatives of the deformed neural field, which are expensive to compute and backpropagate, and result in a rather complex optimization problem (resulting in 14hrs on average, close to the ~10hrs reported by the authors in the GPNF paper). While this formulation is elegant and well-principled, we employ the ARAP formulation (which is a discretization of the elastic energy) by constructing local discretizations of the underlying surface, and achieve continuous propagation over the surface by sampling different patches throughout the optimization. This is not only more efficient to compute and backpropagate, but the optimization converges in significantly fewer iterations, coherently with ARAP for meshes.
>
> **GPFN non-zero volume error**: the neural architecture employed in GPNF is invertible, but it is not volume preserving. The only guarantee coming with invertibility of the deformation is the topological consistency between the source shape and the deformed shape. To achieve volume preservation by construction, more restrictive conditions need to be met: in the supplementary material (A.2.1, Deformation model), we provide the details of the neural architecture we employ, which meets these additional criteria.
>
> **ARAP artefacts**: one of the main advantages of our method (and neural fields-based methods in general) is not being bound to a particular discretization of the input geometry. On the other hand, ARAP and its variants are rather sensitive to triangulation and exhibit these types of artefacts, especially with marching cubes triangulations (which was also previously noted by the authors of GPNF). Furthermore, these mesh-based methods will provide visibly different results for different resolutions of the same geometry, as we show in Figure 17 in the supplementary materials.
>
> **Baseline descriptions**: we agree that the paper would benefit from a paragraph describing the employed baselines, including GPNF. We will include this in the camera-ready revision.

---

> > ### Comment · Reviewer_GZaa · 2025-08-08
> >
> > I thank the authors for providing response to my questions. I have also noted the comments made by other reviewers. I find that my concerns are now satisfied. I would like to raise my score to 5.

---

### Note · Authors · 2025-08-12

Dear AC and Reviewers,

We wish to thank all the reviewers for their work and for engaging in the discussion with us. We are pleased to see that, between our rebuttal and the following interaction, we managed to clear all the concerns that were presented to us in the initial reviews. Following your suggestions, we aim to improve our paper for the camera-ready revision with the following changes:
* **Exposition**
    * **Baselines**: we will dedicate a paragraph in the Experiments section to accurately describing the set of baselines used in our evaluation.
    * **Metrics**: we will move the description of the evaluation metrics, currently located in the supplementary material, to the main text, in the initial part of section 4.
* **References**: we will discuss Neural Jacobian Fields (NJF) and Neural Shape Deformation Priors (NSDP) in our related work section, highlighting their differences wrt our work. NSDP will also be included in our evaluation (see below).
* **Results**
    * **Fig. 5**: we will include an alternate visualization of Figure 5 in the supplementary material.
    * **NSDP**: we will include an evaluation of our method vs NSDP (see rebuttal, reviewer A7SK) in the supplementary materials.
    * **Ground truth**: we will include a quantitative evaluation of our method's approximation of supervised deformation data (namely, the DT4D and DeFAUST datasets), in the supplementary materials.

Best regards,

The authors

---

### Decision · Program_Chairs · 2025-09-17

**Decision:**

Accept (poster)

**Comment:**

The paper received 4 reviews. The authors-reviewers discussion clarified all points raised in the initial reviews. A consensus that the paper makes a useful contribution and can be finalised for publication was reached amongst the reviewers. The paper can be accepted as poster.